# Markov Chain Monte Carlo without Evaluating the Target: an Auxiliary Variable Approach

**Wei Yuan** [1]  **Guanyang Wang** [1]

## Abstract

In sampling tasks, it is common for target distributions to be known up to a normalizing constant. However, in many situations, even evaluating the unnormalized distribution can be costly or infeasible. This issue arises in scenarios such as sampling from the Bayesian posterior for tall datasets and the 'doubly-intractable' distributions. In this paper, we begin by observing that seemingly different Markov chain Monte Carlo (MCMC) algorithms, such as the exchange algorithm, PoissonMH, and TunaMH, can be unified under a simple common procedure. We then extend this procedure into a novel framework that allows the use of auxiliary variables in both the proposal and the acceptance–rejection step. Several new MCMC algorithms emerge from this framework that uses estimated gradients to guide the proposal moves. They have demonstrated significantly better performance than existing methods on both synthetic and real datasets. We also develop theory for the new framework and use it to simplify and extend results for existing algorithms. The code to reproduce the experimental results can be found at https://github.com/ywwes26/Auxiliary-MCMC.

## 1. Introduction

The general workflow of Markov chain Monte Carlo algorithms is alike, yet computational challenges arise in their own way. Interestingly, solutions to these unique challenges sometimes also share a common underlying principle, which often go unnoticed even by their proposers. This paper presents an auxiliary variable-based framework that integrates several MCMC algorithms in Bayesian inference

applications. This unified perspective also inspires the development of novel algorithms with improved performance.

We briefly recall the workflow of Metropolis–Hastings type algorithms (Metropolis et al., 1953; Hastings, 1970). Given a target distribution, these methods typically iterate over a two-step process: they first propose a new state from a proposal distribution given the current state. Then, they apply an acceptance–rejection mechanism to decide if the transition to the new state should occur. Importantly, their implementation often requires evaluating the unnormalized distribution, that is, the target distribution up to a scaling factor. Several algorithms such as the Metropolis–adjusted Langevin algorithm (MALA) and Hamiltonian Monte Carlo (HMC) (Duane et al., 1987) also require evaluating the gradient of the unnormalized distribution.

In Bayesian inference, users focus on quantities associated with the posterior distribution $\pi(\theta \mid x)$, with $x$ as observed data and $\theta$ as the parameter of interest. Given a prior $\pi(\theta)$ and likelihood $p_\theta(x)$, the posterior distribution is $\pi(\theta \mid x) = \pi(\theta)p_\theta(x)/\int p_\theta(x)\pi(\mathrm{d}\theta)$. To 'understand' the typically complex posterior, MCMC is often the go-to strategy. For a detailed historical background, see (Martin et al., 2024).

Although MCMC needs only the unnormalized posterior, which is $\pi(\theta)p_\theta(x)$ as a function of $\theta$ for fixed $x$, computational challenges can still arise from both the $\theta$ and the $x$ side. Consider the following scenarios:

**Scenario 1** (Expensive $\theta$: Doubly-intractable distribution)**.** Suppose the model likelihood $p_\theta(x)$ is expressed as $f_\theta(x)/Z(\theta)$, with $Z(\theta)$ requiring a high-dimensional integration or the summation of an exponential number of terms. Standard Metropolis–Hastings type algorithms cannot be directly applied to these problems, as they necessitate calculating the ratio $Z(\theta')/Z(\theta)$ at each step in its acceptance–rejection process. The posterior distribution $\pi(\theta \mid x) \propto \pi(\theta)f_\theta(x)/Z(\theta)$ is called the "doubly-intractable distribution"(Murray et al., 2006). This distribution has found use in many applications, such as (Robins et al., 2007; Besag, 1986; 1974; Vitelli et al., 2018; Diaconis & Wang, 2018; Strauss, 1975; Hoff, 2009; Rao et al., 2016).

**Scenario 2** (Expensive $x$: Tall Dataset)**.** Imagine that users have gathered a large dataset $x = (x_1, \ldots, x_N)$, where

[1]Department of Statistics, Rutgers University, Piscataway, New Jersey, USA. Correspondence to: Guanyang Wang <guanyang.wang@rutgers.edu>.

*Proceedings of the 43rd International Conference on Machine Learning*, Seoul, South Korea. PMLR 306, 2026. Copyright 2026 by the author(s).

$N$ can be in the millions or billions. Each data point is independently and identically distributed following the distribution $p_\theta$. In this case, the joint model likelihood is $p_\theta(x) = \prod_{i=1}^{N} p_\theta(x_i)$. However, executing standard Metropolis–Hastings algorithms becomes expensive, as the per-step acceptance–rejection computation requires evaluations across the full dataset. This scenario is called "tall data" (Bardenet et al., 2017), which is common in machine learning (Welling & Teh, 2011; Ahn, 2015).

Various approaches have been proposed. In particular, we review three popular algorithms with promising empirical performance: the exchange algorithm for doubly-intractable distributions; PoissonMH (Zhang & De Sa, 2019) and TunaMH (Zhang et al., 2020) for tall datasets. Their details are given in the appendix; see Algorithms 4, 5, and 6 in Appendix A.

Conceptually, all these algorithms are similar. They aim to sample without explicitly calculating the target distribution, while still preserving the correct stationary distribution. However, they vary significantly in their motivations, methodologies, technical assumptions, and algorithmic details. This prompts the question: *Can these algorithms be unified under a common framework?*

### 1.1. Our Contribution

Our first contribution answers the above question affirmatively. We begin with a key observation: the exchange algorithm (Algorithm 4), PoissonMH (Algorithm 5), and TunaMH (Algorithm 6) can be unified into a simple common procedure, detailed in Section 2.

Furthermore, existing methods that avoid evaluating the full posterior mostly rely on random-walk proposals, which mix slowly in high dimensions. Motivated by this, our main contribution is the new framework in Section 3.1, which leads to gradient-based minibatch algorithms that let users use efficient, gradient-informed proposals while preserving the correct stationary distribution under minibatch data.

Our framework (presented in meta-algorithm 1) uses auxiliary variables in both the proposal and the acceptance-rejection step. These auxiliary variables are typically generated in a computationally efficient manner and are used to estimate costly quantities, such as required quantities associated with the proposal and the target ratio.

At a high level, we show that one can design higher-performance MCMC by replacing expensive terms in *both* the proposal and the acceptance–rejection step with cheap estimates, while still preserving the stationary distribution. Our new algorithms (Algorithm 2, 3) support this message and achieve substantially better performance than prior methods and standard MCMC baselines.

Our framework can help remove the systematic bias present in existing minibatch gradient-based MCMC methods that do not converge to the target distribution. An important example is the celebrated stochastic gradient Langevin dynamics (Welling & Teh, 2011) algorithm, which is efficient but exhibit systematic bias, and correcting this bias using minibatch data has remained an open challenge.

**Related works** Our framework is related to pseudo-marginal MCMC, which uses nonnegative unbiased estimators of the unnormalized target to construct exact Metropolis–Hastings chains on an augmented state space (Andrieu & Roberts, 2009). In a standard pseudo-marginal chain, the auxiliary randomness, or equivalently the estimator value, is retained as part of the Markov chain state. By contrast, our framework stores only $\theta$; the auxiliary variables are freshly generated within each transition and are used to form the proposal and the acceptance ratio. This gives a different auxiliary-variable construction from standard pseudo-marginal MCMC.

Another related line studies exact subsampling for piecewise deterministic Markov process (PDMP) samplers, including the Bouncy Particle Sampler (Bouchard-Côté et al., 2018) and the Zig-Zag process (Bierkens et al., 2019). These methods build continuous-time dynamics with the correct stationary distribution and typically avoid full-data evaluations through Poisson thinning, computable bounds, and often control variates. Our method instead gives a discrete-time Metropolis–Hastings construction based on auxiliary variables, leading to different algorithmic trade-offs.

## 2. A Common Procedure of Existing Algorithms

In this section, we give the unifying view described above. The exchange algorithm, PoissonMH, and TunaMH (Algorithm 4, 5, and 6) can all be written as the following auxiliary-variable transition. Given target distribution $\pi(\cdot \mid x)$, a proposal $q(\cdot, \cdot)$, and current state $\theta$:

1. Propose a new state $\theta' \sim q(\theta, \cdot)$
2. Generate an auxiliary variable $\omega$ according to a distribution $P_{\theta \to \theta'}(\cdot)$
3. Construct an estimator $R_{\theta \to \theta'}(\omega)$ for $\pi(\theta' \mid x)/\pi(\theta \mid x)$, set $r := R_{\theta \to \theta'}(\omega)q(\theta', \theta)/q(\theta, \theta')$
4. With probability $\min\{1, r\}$, set $\theta_{\text{new}} := \theta'$. Otherwise, $\theta_{\text{new}} := \theta$.

Compared to standard Metropolis–Hastings algorithms, this method uses an estimator based on an auxiliary variable to estimate the target ratio $\pi(\theta' \mid x)/\pi(\theta \mid x)$ instead of directly computing it. It will also preserve $\pi(\cdot \mid x)$ as the stationary distribution if the following equation is satisfied:

**Proposition 1.** *Let $K$ be the one-step Markov transition kernel defined by the process described above. Suppose for every $\theta, \theta' \in \Theta$, the estimator $R_{\theta \to \theta'}(\omega)$ satisfies*

$$R_{\theta \to \theta'}(\omega)\pi(\theta \mid x)P_{\theta \to \theta'}(\omega) = \pi(\theta' \mid x)P_{\theta' \to \theta}(\omega). \quad (1)$$

*Then $R_{\theta \to \theta'}$ is unbiased for $\pi(\theta' \mid x)/\pi(\theta \mid x)$ and $K$ is reversible with respect to $\pi(\theta \mid x)$.*

Algorithm 4, 5, and 6 all satisfy equation (1) in Proposition 1. Detailed calculations verifying the next three examples are provided in Appendix B.2.

The main new point in this section is the connection we identify between algorithms for doubly intractable distributions and those for tall datasets. The underlying procedure is not new: Section 2.1 of (Andrieu et al., 2018a) gives a slightly more general setup that replaces $P_{\theta' \to \theta}(\omega)$ with $P_{\theta' \to \theta}(\varphi(\omega))$, where $\varphi$ is any measurable involution. Our framework corresponds to the special case $\varphi$ equal to the identity. In contrast, (Andrieu et al., 2018a) focuses on improving estimator quality in the exchange algorithm and does not address tall datasets.

## 3. A New Framework with (Two) Auxiliary Variables

We first introduce our general framework that uses auxiliary variables in both the proposal and the acceptance-rejection mechanism, then discuss how this framework includes previous algorithms as special cases. Furthermore, we develop new gradient-based algorithms inside our framework for the "tall data" scenario based on PoissonMH and TunaMH, with "cheap" estimates of the gradient.

**Motivation: gradient-based proposal design.** The exchange algorithm, PoissonMH, and TunaMH mainly use independent or random-walk proposals, which scale poorly in high dimensions. In contrast, gradient-based MCMC methods such as MALA and HMC are often much more efficient. One simple idea is to choose $q$ in Section 2 as a gradient-based proposal. But this is hard to implement. For doubly intractable distributions, the log-posterior gradient involves the intractable term $\nabla_\theta \log(Z(\theta))$. For tall datasets, computing the gradient requires a full pass over the data, which conflicts with the goal of low per-iteration cost in minibatch methods such as PoissonMH and TunaMH. Nevertheless, designing a gradient-based proposal remains a fascinating idea. Motivated by this, we introduce a new framework in Section 3.1 that extends Section 2 and yields new algorithms that use cheap gradient estimators to guide proposed moves.

### 3.1. General Methodology

We present our method using notations and assumptions in algorithmic and probabilistic terms. We assume that users can access two simulators for generating auxiliary variables, $\mathcal{S}_1$ and $\mathcal{S}_2$. $\mathcal{S}_1$ takes $\theta$ and produces a random variable $\omega_1$. $\mathcal{S}_2$ takes $(\theta, \theta', \omega_1)$ and outputs another random variable $\omega_2$. Formally, consider two measurable spaces $(\Omega_1, \mathcal{F}_1)$ and $(\Omega_2, \mathcal{F}_2)$ with base measures $\lambda_1$ and $\lambda_2$. We define two families of probability measures: $\mathbb{P}_\theta$ on $\Omega_1$ parameterized by $\theta \in \Theta$, and $\mathbb{P}_{\theta, \theta'}(\cdot \mid \omega_1)$ on $\Omega_2$ parameterized by $(\theta, \theta') \in \Theta \times \Theta$ and $\omega_1 \in \Omega_1$. The second family represents the conditional distribution of $\omega_2$ given $\omega_1$ with parameters $(\theta, \theta')$. For any $(\theta, \theta')$, this defines a joint distribution on $\Omega_1 \times \Omega_2$ as $\mathbb{P}_{\theta, \theta'}(\omega_1, \omega_2) = \mathbb{P}_\theta(\omega_1)\mathbb{P}_{\theta, \theta'}(\omega_2 \mid \omega_1)$. Our assumption implies users can simulate from both the marginal distribution of $\omega_1$ and the conditional distribution of $\omega_2$. Additionally, we introduce the single-point probability space NULL := {null}. Setting $\Omega_1$ or $\Omega_2$ to NULL indicates no corresponding auxiliary variable is generated. Lastly, we introduce a family of proposal kernels $\{Q_{\omega_1}\}_{\omega_1 \in \Omega_1}$, where each $\omega_1$ is associated with a Markov transition kernel $Q_{\omega_1}$ on $\Theta$. We assume $Q_{\omega_1}(\theta, \cdot)$ has a density $q_{\omega_1}(\theta, \cdot)$ with respect to a base measure $\lambda$.

The first auxiliary variable $\omega_1$ is used to determine the proposal, with its distribution depending solely on the current state $\theta$. The sampled $\omega_1$ and the current state $\theta$ then determine the proposed state $\theta'$. Then, $\omega_2$, with distribution depending on $\theta, \theta'$, and $\omega_1$, is generated to estimate the target ratio. For instance, $\omega_1$ might be a minibatch uniformly selected from a large dataset, used to estimate the gradient and influence the proposal. An example of $\omega_2$ could be synthetic data generated in the exchange algorithm. Further examples will be presented shortly.

Our meta-algorithm is detailed in Algorithm 1. Algorithm 1 assumes the target is a generic distribution $\Pi$ on $\Theta$. In all applications discussed in this paper, $\Pi(\cdot)$ is the posterior distribution $\pi(\cdot \mid x)$. It is important to note that, evaluating the ratio in Step 6 can be much cheaper than evaluating $\Pi(\theta')/\Pi(\theta_t)$. The costly part of $\Pi(\theta')/\Pi(\theta_t)$ is often offset by $\mathbb{P}_{\theta', \theta_t}(\omega_1, \omega_2)/\mathbb{P}_{\theta_t, \theta'}(\omega_1, \omega_2)$ by design.

The validity of this algorithm can be proven by checking the detailed balance equation:

**Proposition 2.** *Let $\mathbb{P}_{\mathsf{aux}}(\cdot, \cdot)$ be the transition kernel of the Markov chain defined in Algorithm 1. We have $\mathbb{P}_{\mathsf{aux}}(\cdot, \cdot)$ is reversible with respect to $\Pi$.* [1]

From now on, we will always assume $\Pi$ is the posterior

---

[1] The 'acceptance–rejection' mechanism ($\min\{1, r\}$ in Step 7) can be generalized to accept $\theta'$ with a probability of $a(r)$, where $a : [0, \infty) \to [0, 1]$ is any function satisfying $a(t) = ta(1/t)$. The reversibility claimed here still holds. See Appendix B.12 for the proof.

---

**Algorithm 1** Auxiliary-based MCMC

---

1: **Initialize:** Initial state $\theta_0$, auxiliary-based proposal $\{q_{\omega_1}\}_{\omega_1 \in \Omega_1}$; number of iterations $T$; target distribution $\Pi(\theta)$
2: **for** $t = 0$ to $T - 1$ **do**
3:     sample $\omega_1 \sim \mathbb{P}_{\theta_t}(\cdot)$ via $\mathcal{S}_1$
4:     propose $\theta' \sim q_{\omega_1}(\theta_t, \cdot)$
5:     sample $\omega_2 \sim \mathbb{P}_{\theta_t, \theta'}(\cdot \mid \omega_1)$ via $\mathcal{S}_2$
6:     compute the acceptance ratio

$$r \leftarrow \frac{\Pi(\theta')\mathbb{P}_{\theta', \theta_t}(\omega_1, \omega_2)}{\Pi(\theta_t)\mathbb{P}_{\theta_t, \theta'}(\omega_1, \omega_2)} \cdot \frac{q_{w_1}(\theta', \theta_t)}{q_{w_1}(\theta_t, \theta')}$$

7:     with probability $\min\{1, r\}$, set $\theta_{t+1} \leftarrow \theta'$; otherwise, $\theta_{t+1} \leftarrow \theta_t$
8: **end for**

---

$\pi(\cdot \mid x)$. When both $\Omega_1 = \Omega_2 = \mathsf{NULL}$, then no auxiliary variable is generated. Algorithm 1 reduces to the standard Metropolis–Hastings algorithm. We will now briefly discuss several other possibilities.

**Case 1** (Without $\omega_1$). If $\Omega_1 = \mathsf{NULL}$, Algorithm 1 does not use any auxiliary variable for the proposal distribution. The acceptance ratio in Step 6 of Algorithm 1 simplifies to

$$\frac{\pi(\theta' \mid x)\mathbb{P}_{\theta', \theta_t}(\omega_2)}{\pi(\theta_t \mid x)\mathbb{P}_{\theta_t, \theta'}(\omega_2)} \cdot \frac{q(\theta', \theta_t)}{q(\theta_t, \theta')}.$$

This recovers Section 2, which in turn includes the exchange algorithm, PoissonMH, and TunaMH (Algorithm 4, 5, 6).

**Case 2** ($\omega_1 = \omega_2$). Another interesting case is when $\omega_1 = \omega_2$, i.e., the two auxiliary variables are perfectly correlated. In this case, the acceptance ratio reduces to

$$\frac{\pi(\theta' \mid x)\mathbb{P}_{\theta'}(\omega_1)}{\pi(\theta_t \mid x)\mathbb{P}_{\theta_t}(\omega_1)} \cdot \frac{q_{\omega_1}(\theta', \theta_t)}{q_{\omega_1}(\theta_t, \theta')}.$$

In this case, $\omega_1$ can assist in designing the proposal distribution. This approach can sometimes also help estimate the target ratio and reduce per-iteration costs through careful design, which will be discussed in Section 3.2.1. This coincides with the auxiliary Metropolis-Hastings sampler (Section 2.1 of (Titsias & Papaspiliopoulos, 2018)), where the authors introduce an MCMC method that incorporates auxiliary variables in the proposal distribution to enhance sampling efficiency.

**Case 3** ($\omega_1$ independent of $\omega_2$). When $\omega_1$ is independent of $\omega_2$, the acceptance ratio can be written as

$$\frac{\pi(\theta' \mid x)\mathbb{P}_{\theta'}(\omega_1)\mathbb{P}_{\theta', \theta_t}(\omega_2)}{\pi(\theta_t \mid x)\mathbb{P}_{\theta_t}(\omega_1)\mathbb{P}_{\theta_t, \theta'}(\omega_2)} \cdot \frac{q_{\omega_1}(\theta', \theta_t)}{q_{\omega_1}(\theta_t, \theta')}.$$

This approach addresses the issues of 'proposal design' and 'ratio estimation' separately. Users can generate one minibatch to guide the proposal and use the other to estimate the

target ratio. We will discuss this strategy further in Section 3.2.2. The ratio $\mathbb{P}_{\theta'}(\omega_1)/\mathbb{P}_{\theta_t}(\omega_1)$ will also be cancelled out when the distribution of $\omega_1$ does not depend on $\theta$.

Algorithm 1 can also be viewed as an instance of involutive MCMC (Neklyudov et al., 2020). Combine the random variables generated in Steps 3–5 into $v = (\omega_1, \theta', \omega_2)$, with conditional density $m_\theta(v) = P_\theta(\omega_1) q_{\omega_1}(\theta, \theta') P_{\theta, \theta'}(\omega_2 \mid \omega_1)$. The map $f(\theta, \omega_1, \theta', \omega_2) = (\theta', \omega_1, \theta, \omega_2)$ is an involution with unit Jacobian. Under this representation, the involutive-MCMC acceptance ratio reduces exactly to Step 6 of Algorithm 1. Hence the validity proof can be viewed as an application of involutive MCMC validity.

### 3.2. New Algorithms

We now introduce new gradient-based minibatch MCMC algorithms within our framework (Algorithm 1). We focus on sampling from a tall dataset where $\pi(\theta \mid x) \propto \pi(\theta)p_\theta(x) = \pi(\theta) \prod_{i=1}^N \mathsf{p}_\theta(x_i)$. Our goal is to incorporate gradient information in the proposal design while maintaining the correct stationary distribution and minimal overhead. Each iteration only requires a minibatch of data. The high-level idea is to introduce an auxiliary variable ($\omega_1$ in Algorithm 1) to estimate the full gradient at a low cost. Then, we use a second auxiliary variable $\omega_2$, as described in Section 2, to estimate the target ratio.

#### 3.2.1. POISSONMH WITH LOCALLY BALANCED PROPOSAL

In this section, we develop gradient-based PoissonMH. Here, we adhere to all the technical assumptions of PoissonMH (Algorithm 5).

Locally balanced proposals were introduced by (Zanella, 2020) for discrete-state samplers and later developed for continuous spaces by (Livingstone & Zanella, 2022) and (Vogrinc et al., 2023). They provide a useful way to build informed Metropolis–Hastings proposals. We briefly review the Barker's proposal, a gradient-based continuous-space locally balanced proposal that can be more robust than MALA in several settings.

Barker's proposal is of the form $Q^g(\theta, \mathrm{d}\theta') \propto g(\pi(\theta' \mid x)/\pi(\theta \mid x)) K(\theta, \mathrm{d}\theta')$, where $g : \mathbb{R}^+ \to \mathbb{R}^+$ is a *balancing function* satisfying $g(t) = tg(1/t)$. Here, $K$ represents a symmetric kernel. Users can choose the balancing function, with recommended options including $g(t) = \sqrt{t}$ and $g(t) = t/(1+t)$. Experiments on various discrete distributions show that its performance is competitive compared to alternative MCMC methods.

When $\theta$ is in a continuous state space, the proposal $Q^g(\theta, \cdot)$ is not feasible for implementation. However, one can perform a first–order Taylor expansion of $\log(\pi(\theta' \mid x)) -$

$\log(\pi(\theta \mid x))$ with respect to $\theta$, and use the so-called *first-order* locally balanced proposal. It has the form: $Q^{(g)}(\theta, \mathrm{d}\theta') = \prod_{i=1}^{d} Q_i^{(g)}(\theta, \mathrm{d}\theta_i')$. Here $Q_i^{(g)}$ is a one-dimensional kernel of the form

$$Q_i^{(g)}(\theta, \mathrm{d}\theta_i') = Z_i^{-1}(\theta) g \left( e^{\left(\partial_{\theta_i} \log \pi(\theta | x)\right)(\theta_i' - \theta_i)} \right) \cdot \mu_i(\theta_i' - \theta_i) \mathrm{d}\theta_i' \quad (2)$$

where $\mu_i(\cdot)$ represents a symmetric density in $\mathbb{R}$, such as a centred Gaussian. (Livingstone & Zanella, 2022) note that selecting $g(t) = \sqrt{t}$ corresponds to MALA. They suggest using $g(t) = t/(1 + t)$, referred to as 'Barker's proposal', inspired by (Barker, 1965). This choice allows exact calculation of $Z_i(\theta)$, and in turn implies an efficient algorithm for $Q^{(g)}(\theta, \cdot)$. They find that Barker's proposal tends to be more robust compared to MALA.

**Our proposal:** We retain all promises and notations used in Algorithm 5. We also use the notation $\omega_1 := (s_1, s_2, \ldots, s_N)$ and $\mathbb{P}_\theta(\cdot) := \otimes_{i=1}^{N} \mathsf{Poi}\left(\frac{\lambda M_i}{L} + \phi_i(\theta; x)\right)$, as defined in Example 2. For every balancing function $g$, we define the Markov transition kernels with density $q_{\omega_1}^{(g)}(\theta, \theta') := \prod_{i=1}^{d} q_{\omega_1, i}^{(g)}(\theta, \theta_i')$. Here $q_{\omega_1, i}^{(g)}$ is a one-dimensional density proportional to $g\left(e^{\partial_{\theta_i} \log(\pi(\theta | x)\mathbb{P}_\theta(\omega_1))(\theta_i' - \theta_i)}\right) \mu_i(\theta_i' - \theta_i)$, where $\mu_i$ is a symmetric density on $\mathbb{R}$. Our algorithm is summarized in Algorithm 2. Step 3 uses the Poisson minibatch sampling method from (Zhang & De Sa, 2019), enabling faster implementation than sampling each data point independently. See Appendix C for implementation details.

Algorithm 2 corresponds to Case 2 of our meta-algorithm 1. Consequently, its validity is directly proven by Proposition 2. Moreover, several points are worth discussing.

---

**Algorithm 2** Locally Balanced PoissonMH

1: **Initialize:** Initial state $\theta_0$; balancing function $g$; number of iterations $T$
2: **for** $t = 0$ to $T - 1$ **do**
3:      sample $\omega_1 = (s_1, s_2, \cdots, s_N) \sim \mathbb{P}_{\theta_t}(\cdot)$, form minibatch $S = \{i \mid s_i > 0\}$
4:      propose $\theta' \sim q_{\omega_1}^{(g)}(\theta_t, \cdot)$
5:      compute

$$r \leftarrow \frac{\pi(\theta' \mid x)\mathbb{P}_{\theta'}(\omega_1)}{\pi(\theta_t \mid x)\mathbb{P}_{\theta_t}(\omega_1)} \cdot \frac{q_{\omega_1}^{(g)}(\theta', \theta_t)}{q_{\omega_1}^{(g)}(\theta_t, \theta')}.$$

6:      with probability $\min\{1, r\}$, set $\theta_{t+1} \leftarrow \theta'$; otherwise, $\theta_{t+1} \leftarrow \theta_t$
7: **end for**

---

Firstly, the proposed method is an auxiliary-variable-based

variant of the locally-balanced proposal. Our proposal distribution $q_{\omega_1}^{(g)}$ closely resembles the first-order locally balanced proposal, with one key modification. The term $\partial_{\theta_i} \log \pi(\theta \mid x)$ in (2) is substituted by $\partial_{\theta_i} \log (\pi(\theta \mid x)\mathbb{P}_\theta(\omega_1))$. Meanwhile, standard calculation (provided in Appendix D) shows the proxy function $\pi(\theta \mid x) \cdot \mathbb{P}_\theta(\omega_1)$ only depends on **the minibatch** $S$. Consequently, the cost of computing the gradient is at the same level as PoissonMH, thus not significantly increasing the cost per step.

Secondly, following (Livingstone & Zanella, 2022), we choose $g(t) = t/(1 + t)$ and $g(t) = \sqrt{t}$ in our actual implementation. We call them Poisson–Barker and Poisson–MALA respectively. These are the minibatch versions (covering both proposal generation and target ratio evaluation) of the Barker's proposal and MALA. These choices of $g$ allow efficient implementation, with details in Appendix D.

### 3.2.2. TunaMH with SGLD proposal

In this section, we will adhere to all the technical assumptions in TunaMH (Algorithm 6). Recall the target distribution has the form $\pi(\theta \mid x) \propto \exp\left\{-\sum_{i=1}^{N} U_i(\theta; x)\right\}$.

To use the gradient information, we adopt the SGLD algorithm in (Welling & Teh, 2011). At each step, we first select a minibatch $B$ of $K$ data points uniformly from the entire dataset. Thus $(N/K) \sum_{i \in B} \nabla_\theta U_i(\theta_t; x)$ is a natural estimator of the gradient of the negative log-posterior. Calculating this estimator has a cost that scales linearly with the minibatch size $K$, rather than with the total dataset size $N$. Setting $\omega_1 := B \sim \mathsf{Unif}\{\{1, 2, \ldots, N\}, K\}$, our proposal has the form $q_{\omega_1}(\theta, \cdot) \sim \mathbb{N}\left(\theta - \frac{\epsilon^2}{2} \frac{N}{K} \sum_{i \in B} \nabla_\theta U_i(\theta; x), \epsilon^2 \mathbb{I}\right)$, where $\epsilon$ is a tuning parameter. Next, we use the same strategy as TunaMH to select an independent minibatch to estimate the target ratio. Using the notations $\omega_2 := (s_1, s_2, \ldots, s_N)$ and $\mathbb{P}_{\theta, \theta'}(\cdot) := \otimes_{i=1}^{N} \mathsf{Poi}\left(\frac{\lambda c_i}{C} + \phi_i(\theta, \theta'; x)\right)$, our algorithm is summarized in Algorithm 3. Step 5 uses the Poisson minibatch sampling method from (Zhang et al., 2020), enabling faster implementation than sampling each data point independently. See Appendix C for implementation details.

Algorithm 3 corresponds to Case 3 of our meta-algorithm 1. Thus, its validity directly follows from Proposition 2. Our first auxiliary variable $\omega_1$ is independent of the second. Therefore, the distribution $\mathbb{P}_{\theta, \theta'}(\cdot \mid \omega_1) = \mathbb{P}_{\theta, \theta'}(\cdot)$. Additionally, the first variable $\omega_1 = B$ follows a fixed distribution that does not depend on $\theta$. Consequently, the joint distribution $\mathbb{P}_{\theta, \theta'}(\omega_1, \omega_2)$ equals $\mathbb{P}(\omega_1)\mathbb{P}_{\theta, \theta'}(\omega_2)$. As a result, the target-ratio part in Step 6 of Algorithm 1 does not involve the marginal distribution of $\omega_1$.

Algorithm 3 is a Metropolized version of SGLD. If each iteration of Algorithm 3 skips Steps 5–7 for the acceptance–rejection correction, it becomes exactly the SGLD algorithm

described in (Welling & Teh, 2011) with a fixed step size. SGLD is attractive for its computational efficiency, but it is known to suffer from systematic bias. Thus, constructing an acceptance–rejection version of SGLD that uses only minibatch data is viewed as an open problem. To quote from (Welling & Teh, 2011): *"Interesting directions of future research includes deriving a MH rejection step based on minibatch data . . ."* Algorithm 3 offers a solution to this problem.

---

**Algorithm 3** Tuna–SGLD

---

1: **Initialize:** Initial state $\theta_0$; batch size $K$; step size $\epsilon$; number of iterations $T$
2: **for** $t = 0$ to $T - 1$ **do**
3:     sample the first minibatch $B \subset \{1, 2, \cdots, N\}$ uniformly at random with size $K$
4:     propose $\theta' \sim q_{\omega_1}(\theta_t, \cdot)$
5:     sample $\omega_2 = (s_1, s_2, \cdots, s_N) \sim \mathbb{P}_{\theta_t, \theta'}(\cdot)$, form the second minibatch $S = \{i \mid s_i > 0\}$
6:     compute

$$r \leftarrow \frac{\pi(\theta' \mid x) \mathbb{P}_{\theta', \theta_t}(\omega_2)}{\pi(\theta_t \mid x) \mathbb{P}_{\theta_t, \theta'}(\omega_2)} \cdot \frac{q_{\omega_1}(\theta', \theta_t)}{q_{\omega_1}(\theta_t, \theta')}.$$

7:     with probability $\min\{1, r\}$, set $\theta_{t+1} \leftarrow \theta'$; otherwise $\theta_{t+1} \leftarrow \theta_t$
8: **end for**

---

## 4. Theory

We will begin by studying the meta-algorithm 1 and then apply our results to specific algorithms. Our agenda is to compare the Markov chain in Algorithm 1, with transition kernel denoted by $\mathbb{P}_{\mathsf{aux}}$ with two relevant chains $\mathbb{P}_{\mathsf{ideal}}$ and $\mathbb{P}_{\mathsf{MwG}}$. These two chains need not be implementable, but are easier to study as a stochastic process.

Fix $\theta$, define the idealized proposal $Q_{\mathsf{ideal}}(\theta, d\theta') := \mathbb{E}_{\omega_1 \sim \mathbb{P}_\theta}[Q_{\omega_1}(\theta, d\theta')]$ with density $q_{\mathsf{ideal}}(\theta, \theta') = \int_{\Omega_1} q_{\omega_1}(\theta, \theta') \mathbb{P}_\theta(\omega_1) \lambda_1(d\omega_1)$. The idealized chain $\mathbb{P}_{\mathsf{ideal}}$ is the standard Metropolis-Hastings algorithm with proposal $Q_{\mathsf{ideal}}$. Detailed description of $\mathbb{P}_{\mathsf{ideal}}$ is given in Algorithm 7 in Appendix B.4. The second chain, $\mathbb{P}_{\mathsf{MwG}}$, is defined as the transition kernel for Case 2, i.e., $\omega_1 = \omega_2$ in Algorithm 1. A crucial feature is that $\mathbb{P}_{\mathsf{MwG}}$ can equivalently be viewed as a Metropolis–within–Gibbs algorithm. It is the marginal chain for the $\theta$ component, targeting an augmented distribution $\pi(\theta, \omega_1 \mid x) := \pi(\theta \mid x) \mathbb{P}_\theta(\omega_1)$. At each step, users first generate $\omega_1$ from $\pi(\cdot \mid \theta, x) = \mathbb{P}_\theta(\cdot)$. They then use the proposal $q_{\omega_1}(\theta, \cdot)$ to draw $\theta'$, targeting $\pi(\theta' \mid \omega_1, x) \propto \pi(\theta' \mid x) \mathbb{P}_{\theta'}(\omega_1)$, and implement an acceptance–rejection step. This perspective allows us to apply recent results (Qin et al., 2025) in our analysis.

For each $\theta$, we define $p_{\mathsf{ideal}}(\theta, \cdot), p_{\mathsf{MwG}}(\theta, \cdot)$ and $p_{\mathsf{aux}}(\theta, \cdot)$

for the density part of $\mathbb{P}_{\mathsf{ideal}}(\theta, \cdot), \mathbb{P}_{\mathsf{MwG}}(\theta, \cdot)$ and $\mathbb{P}_{\mathsf{aux}}(\theta, \cdot)$.

### 4.1. Peskun's Ordering

The following result generalizes existing results that use only one auxiliary variable, such as Lemma 1 in (Wang, 2022), Section 2.3 in (Nicholls et al., 2012), and Section 2.1 in (Andrieu et al., 2018a). The proof is very similar and is given in Appendix B.5.

**Lemma 1.** *For each pair $(\theta, \theta')$ such that $\theta \neq \theta'$, $p_{\mathsf{aux}}(\theta, \theta') \leq p_{\mathsf{MwG}}(\theta, \theta') \leq p_{\mathsf{ideal}}(\theta, \theta')$.*

This lemma shows that $\mathbb{P}_{\mathsf{aux}}$ is always less likely to move compared to $\mathbb{P}_{\mathsf{ideal}}$ and $\mathbb{P}_{\mathsf{MwG}}$. Lemma 1 implies $\mathbb{P}_{\mathsf{aux}} \prec \mathbb{P}_{\mathsf{MwG}} \prec \mathbb{P}_{\mathsf{ideal}}$ according to Peskun's ordering (Peskun, 1973). This implies that the asymptotic variance for any $f$ under $\mathbb{P}_{\mathsf{ideal}}$ is bounded above by that under $\mathbb{P}_{\mathsf{MwG}}$ and $\mathbb{P}_{\mathsf{aux}}$. For further implications, see (Tierney, 1998).

### 4.2. Comparing the Transition Density

We define the following total variation distance:

$$d_{\mathsf{TV}}(\theta, \theta', \omega_1) := d_{\mathsf{TV}}\left(\mathbb{P}_{\theta, \theta'}(\omega_2 \mid \omega_1), \mathbb{P}_{\theta', \theta}(\omega_2 \mid \omega_1)\right)$$
$$d_{\mathsf{TV}}(\theta, \theta') := \sup_{\omega_1} d_{\mathsf{TV}}(\theta, \theta', \omega_1)$$

Moreover, we also define the following largest KL-divergence:

$$\tilde{d}_{\mathsf{KL}}(\theta, \theta') := \sup_{\omega_1} d_{\mathsf{KL}}\left(\mathbb{P}_{\theta, \theta'}(\omega_2 \mid \omega_1) || \mathbb{P}_{\theta', \theta}(\omega_2 \mid \omega_1)\right)$$

and its symmetrized version:

$$d_{\mathsf{KL}}(\theta, \theta') := \frac{1}{2} \sup_{\omega_1}\left(d_{\mathsf{KL}}(\mathbb{P}_{\theta, \theta'}(\omega_2 \mid \omega_1) || \mathbb{P}_{\theta', \theta}(\omega_2 \mid \omega_1))\right.$$
$$\left. + d_{\mathsf{KL}}(\mathbb{P}_{\theta', \theta}(\omega_2 \mid \omega_1) || \mathbb{P}_{\theta, \theta'}(\omega_2 \mid \omega_1))\right).$$

We have the following comparison result on transition density between $p_{\mathsf{aux}}$ and $p_{\mathsf{MwG}}$.

**Theorem 1.** *For each pair $(\theta, \theta')$ with $\theta \neq \theta'$, we have*

$$p_{\mathsf{aux}}(\theta, \theta') \geq \left(1 - d_{\mathsf{TV}}(\theta, \theta')\right) p_{\mathsf{MwG}}(\theta, \theta')$$
$$\geq e^{-1/e} \exp\left\{-\min\{d_{\mathsf{KL}}(\theta, \theta'), \tilde{d}_{\mathsf{KL}}(\theta, \theta'),\right.$$
$$\left. \tilde{d}_{\mathsf{KL}}(\theta', \theta)\}\right\} p_{\mathsf{MwG}}(\theta, \theta').$$

Theorem 1 also implies comparison on the spectral gaps, as discussed in Theorem 2 in Appendix B.7. It enables us to derive stronger results for specific algorithms, as shown below.

### 4.3. Applications to Existing Algorithms

We will explain how Theorem 1 and Theorem 2 (in Appendix B.7) can be applied to study current algorithms.

**Exchange algorithm:** Since the exchange algorithm (Algorithm 4) has $\omega_1 = $ Null, we know $\mathbb{P}_{\text{MwG}} = \mathbb{P}_{\text{ideal}}$. Our next proposition recovers Theorem 5 in (Wang, 2022).

**Proposition 3.** *Let $\mathbb{P}_{\text{aux}}$ be the transition kernel of Algorithm 4, then we have $p_{\text{aux}}(\theta, \theta') \geq \left(1 - \sup_{\theta, \theta'} d_{\text{TV}}(p_\theta, p_{\theta'})\right) p_{\text{ideal}}(\theta, \theta')$. Let $A_{\theta, \theta'}(s) := \{\omega : p_\theta(\omega) > sp_{\theta'}(\omega)\}$. If there exists $\epsilon, \delta \in (0, 1)$ such that $\mathbb{P}_{p_{\theta'}}(A_{\theta, \theta'}(\delta)) > \epsilon$ uniformly over $\theta, \theta'$, then $\text{Gap}(\mathbb{P}_{\text{aux}}) \geq \epsilon\delta\text{Gap}(\mathbb{P}_{\text{ideal}})$.*

**PoissonMH:** Theorem 2 (in Appendix B.7) allows us to give a (arguably) simpler proof for the theoretical analysis of PoissonMH, with slightly stronger results. The proof is in Appendix B.9.

**Proposition 4.** *With all the notations used in PoissonMH, let $\mathbb{P}_{\text{aux}}$ be the transition kernel of Algorithm 5. Then we have $\text{Gap}(\mathbb{P}_{\text{aux}}) \geq \max\left\{\frac{1}{2}\exp\left(-\frac{L^2}{\lambda+L}\right), e^{-1/e}\exp\left(-\frac{L^2}{2\lambda}\right)\right\}\text{Gap}(\mathbb{P}_{\text{ideal}})$.*

Theorem 2 of Zhang & De Sa (2019) establishes $\text{Gap}(\mathbb{P}_{\text{aux}}) \geq 0.5\exp\left\{\frac{-L^2}{\lambda+L}\right\}\text{Gap}(\mathbb{P}_{\text{ideal}})$ as the main theoretical guarantee for PoissonMH. Since the first term in the maximum is exactly the bound in Zhang & De Sa (2019), our result is never worse, and is strictly sharper if $\lambda > L$.

**TunaMH:** As another application of Theorem 2 (in Appendix B.7), we provide a (again, arguably) simpler proof for TunaMH, achieving stronger results. The proof is in Appendix B.10.

**Proposition 5.** *With all the notations used in TunaMH (Algorithm 6), let $\mathbb{P}_{\text{aux}}$ be the transition kernel of Algorithm 6. Then we have $\text{Gap}(\mathbb{P}_{\text{aux}}) \geq e^{-1/e}\exp\left\{-1/(2\chi)\right\}\text{Gap}(\mathbb{P}_{\text{ideal}})$.*

Theorem 2 of (Zhang et al., 2020) shows $\text{Gap}(\mathbb{P}_{\text{aux}}) \geq \exp\{-1/\chi - 2\sqrt{(\log 2)/\chi}\}\text{Gap}(\mathbb{P}_{\text{ideal}})$. Our rate $\exp\{-1/(2\chi)\}$ is strictly sharper than theirs, with the improvement increasing as $\chi$ decreases.

## 5. Numerical Experiments

We first examine two simulated examples: heterogeneous truncated Gaussian and robust linear regression. We compare our Poisson–Barker and Poisson–MALA algorithms with PoissonMH and full-batch algorithms like random-walk Metropolis, MALA, HMC, Barker (Livingstone & Zanella, 2022), and stochastic gradient MCMC algorithms like SGLD. In the third experiment, we apply Bayesian logistic regression on the `MNIST` dataset, comparing Tuna–SGLD with TunaMH, random-walk Metropolis, MALA, HMC, and Barker. Our results show that our gradient-based minibatch methods significantly outperform existing minibatch and full-batch methods.

In all the figures, the random-walk Metropolis is labeled as MH, while all other methods are labeled by their respective names. Minibatch algorithms require computing several parameters described in Algorithm 5 and 6 before implementation. They are derived in Appendix F.

### 5.1. Heterogeneous Truncated Gaussian

We begin with an illustration example. Suppose $\theta \in \mathbb{R}^d$ is our parameter of interest, the data $\{y_i\}_{i=1}^N$ is generated i.i.d. from $y_i \mid \theta \sim \mathbb{N}(\theta, \Sigma)$ with $\theta = (0, 0, \ldots, 0)$. In our experiment, we set $d = 20$, $N = 10^5$ and $\Sigma = \text{Diagonal}(1, 0.95, 0.90, \cdots, 0.05)$. Therefore, our data follows a multidimensional heterogeneous Gaussian, a common setting in sampling tasks such as (Neal et al., 2011). Following (Zhang & De Sa, 2019; Zhang et al., 2020), we truncate the parameter $\theta$ to $[-3, 3]^d$ and apply a flat prior. As in (Seita et al., 2018; Zhang et al., 2020), we aim to sample from the tempered posterior: $\pi(\theta \mid \{y_i\}_{i=1}^N) \propto \exp\left\{-\frac{1}{2}\beta\sum_{i=1}^N(\theta - y_i)^\intercal\Sigma^{-1}(\theta - y_i)\right\}$ with $\beta = 10^{-5}$.

We test PoissonMH, Poisson–MALA, Poisson–Barker, (full-batch) random-walk Metropolis, (full-batch) MALA, (full-batch) Barker and (minibatch without convergence guarantee) SGLD. Following (Zhang & De Sa, 2019), the hyperparameter $\lambda$ for the three minibatch algorithms is set to $\lambda = 0.0005L^2$, resulting in a batch size of about 6000 (6% of the data points). For SGLD, the minibatch size is set to 512. The initial $\theta_0$ is drawn from $\mathbb{N}(0, \mathbb{I}_d)$, and each algorithm's step size is tuned through several pilot runs to achieve target acceptance rates of 0.25, 0.4, and 0.55. We use Mean Squared Error (MSE) and standard autocorrelation-based Effective Sample Size (ESS) to each dimension of the parameters (identity test function) to compare these methods. We report the minimum, median, and maximum ESS across all dimensions for all methods.

We compare the clock-time performances of the seven methods. Figure 3 shows the MSE over time at different acceptance rates. We find: 1. Poisson–Barker significantly improves PoissonMH, random-walk Metropolis, MALA and Barker for all acceptance rates, both in terms of posterior mean and variance, with more notable improvement at higher acceptance rates. 2. The three exact minibatch algorithms, PoissonMH, Poisson–MALA, and Poisson–Barker, consistently outperform random-walk Metropolis, MALA, and Barker. SGLD decreases MSE quickly at early times but then plateaus, consistent with fixed-step-size bias. 3. Poisson–MALA matches Poisson–Barker's best performance at acceptance rates of 0.4 and 0.55 but shows worse performance than PoissonMH at a 0.25 acceptance rate, aligning with (Livingstone & Zanella, 2022) that MALA is sensitive to tuning, while Barker is more robust. SGLD reduces MSE quickly at early times, likely because it avoids an acceptance–rejection correction. Its curve then plateaus

between 25 and 50 seconds; this behavior is consistent with the fixed-step-size bias of SGLD.

Table 1 compares best ESS per second (ESS/s) across acceptance rates $\{0.25, 0.4, 0.55\}$. Full results are provided in Table 4 in the appendix. Our gradient-based minibatch methods, Poisson–Barker and Poisson–MALA, consistently rank as the top two methods for all acceptance rates (0.25, 0.4, 0.55) and metrics (min, median, max) ESS/s. They improve performance by $1.37 - 7.12$ times over PoissonMH, $4.39 - 9.80$ times over MALA, $6.58 - 15.58$ times over Barker, and $13.62 - 70.12$ times over random-walk Metropolis. Despite the additional gradient estimations, our gradient-based algorithms show substantial benefits from a more efficient proposal.

*Table 1.* ESS/s comparison. (Min, Median, Max) refer to the minimum, median and maximum ESS/s across all dimensions. "Best" reports the best ESS/s across all acceptance rates $\{0.25, 0.4, 0.55\}$. Results are averaged over 100 runs.

| Method | Best ESS/s (Min, Med, Max) |
| --- | --- |
| MH | (0.05, 0.08, 0.47) |
| MALA | (0.10, 0.19, 2.77) |
| Barker | (0.12, 0.22, 1.53) |
| PoissonMH | (0.40, 0.66, 4.67) |
| Poisson–Barker | (**0.91**, **1.65**, 12.16) |
| Poisson–MALA | (0.84, **1.65**, **23.84**) |

### 5.2. Robust Linear Regression

We compare Poisson–Barker, Poisson–MALA, PoissonMH, random-walk Metropolis, MALA, Barker, HMC and SGLD in a robust linear regression example, as considered in (Cornish et al., 2019; Zhang et al., 2020; Maclaurin & Adams, 2014). The data is generated following previous works: for $i = 1, 2, \cdots, N$, covariates $x_i \in \mathbb{R}^d$ are independently generated from $\mathbb{N}(0, \mathbb{I}_d)$ and $y_i = \sum_{j=1}^{d} x_{ij} + \epsilon_i$ with $\epsilon_i \sim \mathbb{N}(0, 1)$. The likelihood is modeled as $p(y_i \mid \theta, x_i) = \text{Student}(y_i - \theta^\intercal x_i \mid v)$, where $\text{Student}(\cdot \mid v)$ is the density of a Student's t distribution with $v$ degrees of freedom. The model is "robust" due to the heavier tail of the likelihood function (Maclaurin & Adams, 2014). With a flat prior, the tempered posterior is: $\pi(\theta \mid \{y_i, x_i\}_{i=1}^{N}) \propto \exp\left\{-\beta \cdot \frac{v+1}{2} \sum_{i=1}^{N} \log\left(1 + \frac{(y_i - \theta^\intercal x_i)^2}{v}\right)\right\}$. We also follow the common practice (Gelman et al., 2008) of constraining the coefficients within a high-dimensional sphere $\{\theta \in \mathbb{R}^d \mid \|\theta\|_2 \leq R\}$.

We use $v = 4$ as in (Cornish et al., 2019; Zhang et al., 2020) and set $d = 10$, $N = 10^5$, $\beta = 10^{-4}$, $R = 15$. PoissonMH, Poisson–MALA and Poisson–Barker use $\lambda = 0.01L^2$. We compare the performance of HMC under different choices of leapfrog steps and set it to 10; see Section G.2 in the appendix for details. The initial $\theta_0$ is drawn from $\mathbb{N}(0, \mathbb{I}_d)$,

and the step size of each algorithm is tuned to achieve acceptance rates of $0.25, 0.4$, and $0.55$. Our true parameter is $\theta^\star = (1, 1, \ldots, 1)$.

Figure 1 shows the MSE of estimating the true coefficients over time. Poisson–MALA and Poisson–Barker are much more efficient than PoissonMH, and all minibatch methods outperform full-batch methods. We also present the best ESS/s comparisons in Table 2. Full results are provided in Table 5 in the appendix. The results have a similar trend to Section 5.1. Poisson–MALA and Poisson–Barker show improvements ranging from 1.80 to 8.79 times over PoissonMH, and achieving nearly or more than 100 times the performance of full-batch methods.

Appendices G.2 and G.3 provide additional experiments covering different parameter settings, higher dimensions and larger data size.

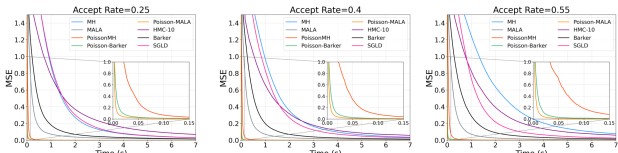

*Figure 1.* MSE of estimating $\theta^\star$ as a function of time across different acceptance rates. The three large plots show the performance of all methods in the first 7 seconds. The three inside plots zoom in on PoissonMH, Poisson–Barker, and Poisson–MALA in the first 0.5 seconds. All results are averaged over 100 runs.

*Table 2.* ESS/s comparison. (Min, Median, Max) denote the minimum, median and maximum ESS/s across all dimensions. "Best" reports the best ESS/s over acceptance rates $\{0.25, 0.4, 0.55\}$. Results are averaged over 100 runs.

| Method | Best ESS/s (Min, Med, Max) |
| --- | --- |
| MH | (2.0, 2.1, 2.1) |
| MALA | (5.0, 5.1, 5.2) |
| Barker | (2.9, 3.0, 3.0) |
| HMC-10 | (0.9, 0.9, 1.0) |
| PoissonMH | (99.0, 100.2, 101.1) |
| Poisson–Barker | (265.3, 268.3, 270.3) |
| Poisson–MALA | (**489.3**, **491.8**, **496.5**) |

### 5.3. Bayesian Logistic Regression

We compare Tuna–SGLD, TunaMH, MALA, Barker, HMC, random-walk Metropolis on a Bayesian logistic regression task using the `MNIST` handwritten digits dataset. We focus on classifying handwritten 3s and 5s, which can often appear visually similar. The training and test set have 11,552 and 1,902 samples, respectively. Following (Welling & Teh, 2011; Zhang et al., 2020), we use the first 50 principal components of the $28 \times 28$ features as covariates. For TunaMH and Tuna–SGLD, we follow (Zhang et al., 2020) and set

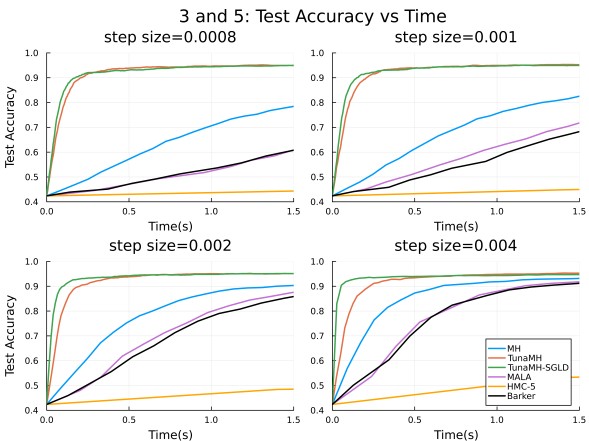

*Figure 2.* Test accuracy as a function of time. Tuna–SGLD is our proposed method.

the hyperparameter $\chi = 10^{-5}$. The batch size $|B|$ in Tuna–SGLD is set as 20, since it is sufficient for Tuna–SGLD to converge fast and using larger minibatch sizes will bring additional computational cost; see Section G.4. We vary step sizes in $0.0008, 0.001, 0.002, 0.004$ for all algorithms. HMC uses 5 leapfrog steps.

The test accuracy comparisons are shown in Figure 2. Due to an extra minibatch for gradient estimation, Tuna–SGLD converges faster than TunaMH for all step sizes, especially larger ones. Both of them significantly outperform full-batch methods, consistent with our previous observations. HMC converges poorly with the step sizes considered.

We test all methods across a wider range of hyperparameter choices in appendix G.4, and provide experimental results on classifying 7s and 9s in appendix G.5.

### 5.4. MCMC Correctness Checks

We add an empirical correctness check for PoissonMH, Poisson–MALA, Poisson–Barker, random-walk Metropolis, MALA, Barker, and SGLD. We use the truncated Gaussian example from Section 5.1 and generate reference posterior samples using rejection sampling. Under a matched computation budget, we discard the first 20% of iterations as burn-in and compute two-sample Kolmogorov–Smirnov (KS) statistics against the reference samples. Table 3 shows that Poisson–Barker and Poisson–MALA have maximum KS statistics no larger than 0.05 in this experiment. In contrast, SGLD reaches a maximum KS statistic of 0.18, consistent with the known fixed-step-size bias of SGLD. These results support that our methods are sampling from the correct target distribution in this benchmark.

*Table 3.* Kolmogorov–Smirnov statistics comparison with different target accept rates. (Min, Median, Max) denote the minimum, median and maximum across all dimensions. Note that SGLD does not have a target accept rate.

| | KS statistics: (Min, Median, Max) | | |
|---|---|---|---|
| Method | target rate=0.25 | target rate=0.4 | target rate=0.55 |
| MH | (0.03, 0.07, 0.18) | (0.03, 0.07, 0.18) | (0.03, 0.08, 0.18) |
| MALA | (0.04, 0.06, 0.15) | (0.03, 0.06, 0.13) | (0.03, 0.05, 0.11) |
| Barker | (0.03, 0.05, 0.10) | (0.02, 0.04, 0.10) | (0.02, 0.04, 0.09) |
| SGLD | same | (0.06, 0.12, 0.18) | same |
| PoissonMH | (0.01, 0.02, 0.06) | (0.01, 0.02, 0.06) | (0.01, 0.03, 0.08) |
| Poisson–Barker | (0.01, 0.02, 0.05) | (0.01, 0.02, 0.05) | (0.01, 0.02, 0.04) |
| Poisson–MALA | (0.01, 0.03, 0.05) | (0.01, 0.02, 0.04) | (0.01, 0.02, 0.04) |

## 6. Future Directions

The proposed algorithms have several limitations. First, they require precomputed constants or bounds, such as $M_i, c_i, L$ and $C$; in some models these quantities may be unavailable or too loose to be useful. Second, their performance depends on hyperparameters that control minibatch size and estimator variance. Third, the current algorithms are designed for independent-data likelihoods.

These limitations naturally suggest two future directions. First, minibatch MCMC should be developed for non-i.i.d. models. For i.i.d. data, per-iteration cost is roughly proportional to minibatch size (e.g., $5\%$ data $\approx 5\%$ cost). In contrast, many non-i.i.d. posteriors are much more expensive to evaluate (often quadratic or cubic in sample size), so a $5\%$ minibatch may cost $\approx 0.25\%$ (or less) of a full-batch step. This makes non-i.i.d. settings such as spatial models and independent component analysis particularly important. Second, the source of empirical speedups remains unclear: some minibatch methods report $10$–$100\times$ higher ESS/s than full-data MCMC, yet (Johndrow et al., 2020) shows that several minibatch schemes (including (Maclaurin & Adams, 2014)) do not yield meaningful gains for generalized linear models. One possible link is whether a small subset can approximate the full posterior, as in coreset methods (Huggins et al., 2016; Campbell & Broderick, 2018; 2019).

## Acknowledgement

The authors are partially supported by the National Science Foundation through grant DMS-2210849 and an Adobe Data Science Research Award. The authors thank Qian Qin and Pierre Jacob for helpful discussions. The authors thank the four anonymous reviewers for their insightful suggestions during the rebuttal process.

## Impact Statement

This paper presents work whose goal is to advance the field of machine learning. There are many potential societal consequences of our work, none of which we feel must be

specifically highlighted here.

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

## A. Details of Exchange algorithm, PoissonMH, TunaMH

The beginning of each algorithm's pseudocode contains a section titled "promise," which describes the technical assumptions and practical prerequisites that users need to know before executing the algorithm. Two points merit attention. First, the posterior with $N$ i.i.d. data points satisfies this promise of PoissonMH (and TunaMH) by setting $\phi_i(\theta, x) := N^{-1} \log \pi(\theta) + \log \mathsf{p}_\theta(x_i)$ (and $U_i(\theta, x) := -N^{-1} \log \pi(\theta) - \log \mathsf{p}_\theta(x_i)$.) Second, the naïve implementation of the Poisson minibatch sampling (Step 5 in Algorithm 5, Step 6 in Algorithm 6) has a cost linear with the total data size, since each data point is associated with a Poisson variable. However, as highlighted in (Zhang & De Sa, 2019; Zhang et al., 2020), the 'Poisson thinning' trick can be used to implement Poisson minibatch sampling much more efficiently. Therefore, its per-step cost is much lower than both the naïve implementation and full-batch methods. See Section C in Appendix for implementation details and proofs.

---

**Algorithm 4** Exchange Algorithm

---

1: **Promise:**
   - Target $\pi(\theta \mid x) \propto \pi(\theta) p_\theta(x)$,
   - Likelihood $p_\theta(x) = f_\theta(x)/Z(\theta)$ where the value of $f_\theta$ can be queried
   - A simulator $\mathcal{S}(\cdot)$ with input $\theta$. Each run outputs a random variable $\omega \sim p_\theta$

2: **Initialize:** Initial state $\theta_0$, proposal $q$; number of iterations $T$
3: **for** $t = 0$ to $T - 1$ **do**
4:     propose $\theta' \sim q(\theta_t, \cdot)$
5:     sample $\omega \sim \mathcal{S}(\theta')$
6:     compute

$$r \leftarrow \frac{\pi(\theta') f_{\theta'}(x) f_{\theta_t}(\omega)}{\pi(\theta_t) f_{\theta_t}(x) f_{\theta'}(\omega)} \cdot \frac{q(\theta', \theta_t)}{q(\theta_t, \theta')}$$

7:     with probability $\min\{1, r\}$, set $\theta_{t+1} \leftarrow \theta'$; otherwise, $\theta_{t+1} \leftarrow \theta_t$
8: **end for**

---

---

**Algorithm 5** PoissonMH

---

1: **Promise:**
   - Target $\pi(\theta \mid x) \propto \exp\left\{\sum_{i=1}^N \phi_i(\theta; x)\right\}$,
   - For every $\theta$, each $\phi_i(\theta; x) \in [0, M_i]$ with some $M_i > 0$

2: **Initialize:** Initial state $\theta_0$; proposal $q$; hyperparameter $\lambda$; set $L := \sum_{i=1}^N M_i$; number of iterations $T$
3: **for** $t = 0$ to $T - 1$ **do**
4:     propose $\theta' \sim q(\theta_t, \cdot)$
5:     sample $s_i \sim \mathsf{Poi}\left(\frac{\lambda M_i}{L} + \phi_i(\theta_t; x)\right)$ for each $i \in \{1, 2, \ldots, N\}$, form minibatch $S = \{i \mid s_i > 0\}$
6:     compute

$$r \leftarrow \frac{\exp\left\{\sum_{i \in S} s_i \log\left(1 + \frac{L}{\lambda M_i} \phi_i(\theta'; x)\right)\right\}}{\exp\left\{\sum_{i \in S} s_i \log\left(1 + \frac{L}{\lambda M_i} \phi_i(\theta_t; x)\right)\right\}} \cdot \frac{q(\theta', \theta_t)}{q(\theta_t, \theta')}$$

7:     with probability $\min\{1, r\}$, set $\theta_{t+1} \leftarrow \theta'$; otherwise $\theta_{t+1} \leftarrow \theta_t$
8: **end for**

---

---
**Algorithm 6** TunaMH
---

1: **Promise:**
   - Target $\pi(\theta \mid x) \propto \exp\left\{-\sum_{i=1}^{N} U_i(\theta; x)\right\}$,
   - For every $\theta$ and $\theta'$, each $|U_i(\theta'; x) - U_i(\theta; x)| \leq c_i M(\theta, \theta')$ for some symmetric non-negative function $M$ and positive constants $c_i$

2: **Initialize:** Initial state $\theta_0$; proposal $q$; hyperparameter $\chi$; set $C = \sum_{i=1}^{N} c_i$; number of iterations $T$
3: **for** $t = 0$ to $T - 1$ **do**
4:     propose $\theta' \sim q(\theta_t, \cdot)$ and compute $M(\theta_t, \theta')$
5:     set $\lambda = \chi C^2 M^2(\theta_t, \theta')$
6:     sample $s_i \sim \text{Poi}(\frac{\lambda c_i}{C} + \phi_i(\theta_t, \theta'; x))$ where $\phi_i(\theta_t, \theta'; x) = \frac{U_i(\theta'; x) - U_i(\theta_t; x)}{2} + \frac{c_i}{2} M(\theta_t, \theta')$, for each $i \in \{1, 2, \cdots, N\}$, form minibatch $S = \{i \mid s_i > 0\}$
7:     compute

$$r \leftarrow \frac{\exp\left\{\sum_{i \in S} s_i \log\left(1 + \frac{C}{\lambda c_i} \phi_i(\theta', \theta_t; x)\right)\right\}}{\exp\left\{\sum_{i \in S} s_i \log\left(1 + \frac{C}{\lambda c_i} \phi_i(\theta_t, \theta'; x)\right)\right\}} \cdot \frac{q(\theta', \theta_t)}{q(\theta_t, \theta')}$$

8:     with probability $\min\{1, r\}$, set $\theta_{t+1} \leftarrow \theta'$; otherwise $\theta_{t+1} \leftarrow \theta_t$
9: **end for**

---

# B. Additional Proofs

## B.1. Proof of Proposition 1

*Proof of Proposition 1.* For unbiasedness, integrating both sides of (1) with respect to $\omega$ gives:

$$\pi(\theta \mid x)\mathbb{E}_{P_{\theta \to \theta'}}[R_{\theta \to \theta'}(\omega)] = \pi(\theta' \mid x) \int P_{\theta' \to \theta}(\omega)\mathrm{d}\omega = \pi(\theta' \mid x),$$

where the second equality follows from the fact that integrating a probability measure over the entire space equals one. Dividing both sides by $\pi(\theta \mid x)$ gives us the desired result.

For reversibility, we prove this by checking the detailed balance equation. For every fixed $\theta \neq \theta'$, we aim to show:

$$\pi(\theta \mid x)K(\theta, \theta') = \pi(\theta' \mid x)K(\theta', \theta)$$

Expanding the left side of the above equation:

$$
\begin{aligned}
\pi(\theta \mid x)K(\theta, \theta') &= \pi(\theta \mid x)q(\theta, \theta')\mathbb{E}_\omega\left[\min\left(R_{\theta \to \theta'}(\omega)\frac{q(\theta', \theta)}{q(\theta, \theta')}, 1\right)\right] \\
&= \int_\Omega \pi(\theta \mid x)q(\theta, \theta')P_{\theta \to \theta'}(\omega)\min\left(R_{\theta \to \theta'}(\omega)\frac{q(\theta', \theta)}{q(\theta, \theta')}, 1\right)\mathrm{d}\omega \\
&= \int_\Omega \min\left(\pi(\theta \mid x)P_{\theta \to \theta'}(\omega)R_{\theta \to \theta'}(\omega)q(\theta', \theta), \pi(\theta \mid x)q(\theta, \theta')P_{\theta \to \theta'}(\omega)\right)\mathrm{d}\omega \\
&= \int_\Omega \min\left(\pi(\theta' \mid x)P_{\theta' \to \theta}(\omega)q(\theta', \theta), \pi(\theta \mid x)q(\theta, \theta')P_{\theta \to \theta'}(\omega)\right)\mathrm{d}\omega \qquad \text{by (1)} \\
&= \int_\Omega \min\left(\pi(\theta' \mid x)P_{\theta' \to \theta}(\omega)q(\theta', \theta), q(\theta, \theta')R_{\theta' \to \theta}(\omega)\pi(\theta' \mid x)P_{\theta' \to \theta}(\omega)\right)\mathrm{d}\omega \qquad \text{by (1)} \\
&= \int_\Omega \pi(\theta' \mid x)P_{\theta' \to \theta}(\omega)q(\theta', \theta)\min\left(1, \frac{q(\theta, \theta')}{q(\theta', \theta)}R_{\theta' \to \theta}(\omega)\right)\mathrm{d}\omega \\
&= \pi(\theta' \mid x)q(\theta', \theta)\mathbb{E}_\omega\left[\min\left(1, \frac{q(\theta, \theta')}{q(\theta', \theta)}R_{\theta' \to \theta}(\omega)\right)\right] \\
&= \pi(\theta' \mid x)K(\theta', \theta),
\end{aligned}
$$

gives the desired result. $\square$

## B.2. Verification of Examples in Section 2

**Example 1** (Exchange algorithm). Exchange algorithm (Algorithm 4) aims to sample from $\pi(\theta|x) \propto p(\theta)\frac{f_\theta(x)}{Z(\theta)}$ for doubly-intractable distributions. The auxiliary variable $\omega$ is an element in the sample space. Meanwhile:

- $P_{\theta \to \theta'}(\omega) := p_{\theta'}(\omega) = f_{\theta'}(\omega)/Z(\theta')$.

- The estimator

$$R_{\theta \to \theta'}(\omega) := \frac{\pi(\theta')f_{\theta'}(x)f_\theta(\omega)}{\pi(\theta)f_\theta(x)f_{\theta'}(\omega)}.$$

**Example 2** (PoissonMH). PoissonMH (Algorithm 5) aims to sample from $\pi(\theta \mid x) \propto \exp(\sum_{i=1}^N \phi_i(\theta; x))$. The auxiliary variable $\omega = (s_1, s_2, \cdots, s_N)$ belongs to $\{0, 1, 2, \cdots\}^N$, i.e., a vector of natural numbers with the same length as the number of data points. Each component of $\omega$ follows an independent Poisson distribution. More precisely:

- $P_{\theta \to \theta'} := \bigotimes_{i=1}^N \text{Poi}\left(\frac{\lambda M_i}{L} + \phi_i(\theta; x)\right)$.

- The estimator

$$R_{\theta \to \theta'}(\omega) := \frac{\exp\left\{\sum_{i \in S} s_i \log\left(1 + \frac{L}{\lambda M_i}\phi_i(\theta'; x)\right)\right\}}{\exp\left\{\sum_{i \in S} s_i \log\left(1 + \frac{L}{\lambda M_i}\phi_i(\theta; x)\right)\right\}},$$

with all the notations defined in Algorithm 5. The estimator relies solely on data points in $S$. By selecting appropriate parameters, the expected size of $S$ can be much smaller than $N$, thereby reducing the computational cost per iteration.

**Example 3** (TunaMH). TunaMH (Algorithm 6) aims to solve the same problem as PoissonMH under more practical assumptions. Similar to PoissonMH, the auxiliary variable $\omega = (s_1, s_2, \cdots, s_N) \in \{0, 1, 2, \cdots\}^N$. Meanwhile:

- The $P_{\theta \to \theta'} := \bigotimes_{i=1}^N \text{Poi}\left(\frac{\lambda c_i}{C} + \phi_i(\theta, \theta'; x)\right)$.

- The estimator

$$R_{\theta \to \theta'}(\omega) := \frac{\exp\left\{\sum_{i \in S} s_i \log\left(1 + \frac{C}{\lambda c_i}\phi_i(\theta', \theta; x)\right)\right\}}{\exp\left\{\sum_{i \in S} s_i \log\left(1 + \frac{C}{\lambda c_i}\phi_i(\theta, \theta'; x)\right)\right\}}.$$

with all the notations defined in Algorithm 6.

### B.2.1. VERIFICATION OF EXAMPLE 1

To check equation (1), recall $\pi(x) := \int \pi(\theta)p_\theta(x)d\theta$ is the marginal distribution of $x$:

$$
\begin{aligned}
R_{\theta \to \theta'}(\omega) \cdot \pi(\theta \mid x) \cdot P_{\theta \to \theta'}(\omega) &= \frac{\pi(\theta')f_{\theta'}(x)f_\theta(\omega)}{\pi(\theta)f_\theta(x)f_{\theta'}(\omega)} \cdot \frac{\pi(\theta)f_\theta(x)}{Z(\theta)\pi(x)} \cdot \frac{f_{\theta'}(\omega)}{Z(\theta')} \\
&= \frac{\pi(\theta')f_{\theta'}(x)f_\theta(\omega)}{\pi(\theta)f_\theta(x)f_{\theta'}(\omega)} \cdot \frac{\pi(\theta)f_\theta(x)}{Z(\theta)\pi(x)} \cdot \frac{f_{\theta'}(\omega)}{Z(\theta')} \\
&= \frac{\pi(\theta')f_{\theta'}(x)}{\pi(x)Z(\theta')} \cdot \frac{f_\theta(\omega)}{Z(\theta)} \\
&= \pi(\theta' \mid x)p_\theta(\omega) \\
&= \pi(\theta' \mid x)P_{\theta' \to \theta}(\omega).
\end{aligned}
$$

All the equalities follow from either the Bayes formula, or the problem definition.

B.2.2. VERIFICATION OF EXAMPLE 2

To check equation (1), recall $\pi(\theta \mid x) = \exp(\sum_{i=1}^{N} \phi_i(\theta; x))/Z(x)$ :

$$R_{\theta \to \theta'}(\omega) \cdot \pi(\theta \mid x) \cdot P_{\theta \to \theta'}(\omega) = R_{\theta \to \theta'}(\omega) \frac{\exp(\sum_{i=1}^{N} \phi_i(\theta; x))}{Z(x)} \prod_{i=1}^{N} P_{\theta \to \theta'}(s_i)$$

$$= \frac{1}{Z(x)} \prod_{i=1}^{N} \left( \frac{\lambda M_i L^{-1} + \phi_i(\theta'; x)}{\lambda M_i L^{-1} + \phi_i(\theta; x)} \right)^{s_i} \cdot e^{\phi_i(\theta;x)} \cdot e^{-\lambda M_i L^{-1}} e^{-\phi_i(\theta;x)} \frac{(\lambda M_i L^{-1} + \phi_i(\theta; x))^{s_i}}{s_i!}$$

$$= \frac{1}{Z(x)} \prod_{i=1}^{N} \left( \frac{\lambda M_i L^{-1} + \phi_i(\theta'; x)}{\cancel{\lambda M_i L^{-1} + \phi_i(\theta; x)}} \right)^{s_i} \cdot \cancel{e^{\phi_i(\theta;x)}} \cdot e^{-\lambda M_i L^{-1}} \cancel{e^{-\phi_i(\theta;x)}} \frac{\left( \cancel{\lambda M_i L^{-1} + \phi_i(\theta; x)} \right)^{s_i}}{s_i!}$$

$$= \frac{1}{Z(x)} \prod_{i=1}^{N} \frac{(\lambda M_i L^{-1} + \phi_i(\theta'; x))^{s_i}}{s_i!} (e^{-\lambda M_i L^{-1}} e^{-\phi_i(\theta';x)}) e^{\phi_i(\theta';x)}$$

$$= \pi(\theta' \mid x) \prod_{i=1}^{N} P_{\theta' \to \theta}(s_i)$$

$$= \pi(\theta' \mid x) P_{\theta' \to \theta}(\omega).$$

B.2.3. VERIFICATION OF EXAMPLE 3

To verify equation (1), first recall $\pi(\theta \mid x) = \exp\left( -\sum_{i=1}^{N} U_i(\theta; x) \right)/Z(x)$, and the definition

$$\phi_i(\theta, \theta'; x) = \frac{U_i(\theta'; x) - U_i(\theta; x)}{2} + \frac{c_i}{2} M(\theta, \theta').$$

It is useful to observe:

$$U_i(\theta; x) + \phi(\theta, \theta'; x) = \frac{U_i(\theta'; x) + U_i(\theta; x)}{2} + \frac{c_i M(\theta, \theta')}{2} = U_i(\theta'; x) + \phi_i(\theta', \theta; x).$$

We have :

$$R_{\theta \to \theta'}(\omega) \cdot \pi(\theta \mid x) \cdot P_{\theta \to \theta'}(\omega) = R_{\theta \to \theta'}(\omega) \frac{\exp\left( -\sum_{i=1}^{N} U_i(\theta; x) \right)}{Z(x)} \prod_{i=1}^{N} P_{\theta \to \theta'}(s_i)$$

$$= \frac{1}{Z(x)} \prod_{i=1}^{N} \left( \frac{\lambda c_i C^{-1} + \phi_i(\theta', \theta; x)}{\lambda c_i C^{-1} + \phi_i(\theta, \theta'; x)} \right)^{s_i} \cdot e^{-\phi_i(\theta,\theta';x)} \cdot e^{-\lambda c_i C^{-1}} e^{-U_i(\theta;x)} \frac{(\lambda c_i C^{-1} + \phi_i(\theta, \theta'; x))^{s_i}}{s_i!}$$

$$= \frac{1}{Z(x)} \prod_{i=1}^{N} \left( \frac{\lambda c_i C^{-1} + \phi_i(\theta', \theta; x)}{\cancel{\lambda c_i C^{-1} + \phi_i(\theta, \theta'; x)}} \right)^{s_i} \cdot e^{-\phi_i(\theta',\theta;x)} \cdot e^{-\lambda c_i C^{-1}} e^{-U_i(\theta';x)} \frac{\left( \cancel{\lambda c_i C^{-1} + \phi_i(\theta, \theta'; x)} \right)^{s_i}}{s_i!}$$

$$= \frac{1}{Z(x)} \prod_{i=1}^{N} \frac{(\lambda c_i C^{-1} + \phi_i(\theta', \theta; x))^{s_i}}{s_i!} (e^{-\lambda c_i C^{-1}} e^{-\phi_i(\theta',\theta;x)}) e^{-U_i(\theta';x)}$$

$$= \pi(\theta' \mid x) \prod_{i=1}^{N} P_{\theta' \to \theta}(s_i)$$

$$= \pi(\theta' \mid x) P_{\theta' \to \theta}(\omega).$$

## B.3. Proof of Proposition 2

*Proof of Proposition 2.* For each $\theta \in \Theta$, the probability distribution $\mathbb{P}_{\mathsf{aux}}(\theta, \cdot)$ can be decomposed as a point-mass at $\theta$ and a density on $\Theta$, namely:

$$\mathbb{P}_{\mathsf{aux}}(\theta, \mathrm{d}\theta') = R(\theta)\delta_\theta(\mathrm{d}\theta') + p_{\mathsf{aux}}(\theta, \theta')\lambda(\mathrm{d}\theta'),$$

where $\lambda$ is the base measure on $\Theta$ defined earlier. Our goal is to show $\Pi(\theta)p_{\mathsf{aux}}(\theta, \theta') = \Pi(\theta')p_{\mathsf{aux}}(\theta', \theta)$ for every pair $(\theta, \theta')$ satisfying $\theta \neq \theta'$. Reversibility will directly result from this equality. The term $p_{\mathsf{aux}}(\theta, \theta')$ describes the density of successfully moving from $\theta$ to $\theta'$, including two steps: proposing $\theta'$ and then accepting the transition. Therefore we have the following expression of $p_{\mathsf{aux}}(\theta, \theta')$:

$$p_{\mathsf{aux}}(\theta, \theta') =$$
$$\int_{\Omega_1} \mathbb{P}_\theta(\omega_1) q_{\omega_1}(\theta, \theta') \int_{\Omega_2} \left( \min \left\{ 1, \frac{\Pi(\theta')\mathbb{P}_{\theta',\theta}(\omega_1, \omega_2)}{\Pi(\theta)\mathbb{P}_{\theta,\theta'}(\omega_1, \omega_2)} \cdot \frac{q_{w_1}(\theta', \theta)}{q_{w_1}(\theta, \theta')} \right\} \mathbb{P}_{\theta,\theta'}(\omega_2 \mid \omega_1) \lambda_2(\mathrm{d}\omega_2) \right) \lambda_1(\mathrm{d}\omega_1).$$

Therefore we have

$$\Pi(\theta)p_{\mathsf{aux}}(\theta, \theta') =$$
$$\int \min \left\{ \Pi(\theta)\mathbb{P}_{\theta,\theta'}(\omega_1, \omega_2) q_{\omega_1}(\theta, \theta'), \Pi(\theta')\mathbb{P}_{\theta',\theta}(\omega_1, \omega_2) q_{w_1}(\theta', \theta) \right\} \lambda_1 \otimes \lambda_2(\mathrm{d}\omega_1 \mathrm{d}\omega_2),$$

which is symmetric over $\theta$ and $\theta'$. This shows $\Pi(\theta)p_{\mathsf{aux}}(\theta, \theta') = \Pi(\theta')p_{\mathsf{aux}}(\theta', \theta)$. $\qquad\square$

### B.4. Algorithmic Description of the Idealized Chain

---
**Algorithm 7** idealized MH

---
1: **Initialize:** Initial state $\theta_0$; number of iterations $T$; target distribution $\pi(\theta \mid x)$
2: **for** $t = 0$ to $T - 1$ **do**
3:      propose $\theta' \sim Q_{\mathsf{ideal}}(\theta_t, \cdot)$
4:      compute the acceptance ratio

$$r \leftarrow \frac{\pi(\theta' \mid x)}{\pi(\theta_t \mid x)} \cdot \frac{q_{\mathsf{ideal}}(\theta', \theta_t)}{q_{\mathsf{ideal}}(\theta_t, \theta')}$$

5:      with probability $\min\{1, r\}$, set $\theta_{t+1} \leftarrow \theta'$; otherwise, $\theta_{t+1} \leftarrow \theta_t$
6: **end for**

---

### B.5. Proof of Lemma 1

*Proof of Lemma 1.* We write $p_{\mathsf{aux}}(\theta, \theta')$ in its integral form, and upper bound it as follows:

$$p_{\mathsf{aux}}(\theta, \theta') = \int_{\Omega_1 \times \Omega_2} \mathbb{P}_\theta(\omega_1) q_{\omega_1}(\theta, \theta') \mathbb{P}_{\theta,\theta'}(\omega_2 \mid \omega_1) r(\theta, \theta'; \omega_1, \omega_2) \lambda_1 \otimes \lambda_2(\mathrm{d}\omega_1 \mathrm{d}\omega_2)$$

$$= \int_{\Omega_1} \mathbb{P}_\theta(\omega_1) q_{\omega_1}(\theta, \theta') \left( \mathbb{E}_{\omega_2 \mid \omega_1, \theta, \theta'} \left[ \min \left\{ \frac{\pi(\theta' \mid x)\mathbb{P}_{\theta',\theta}(\omega_1, \omega_2)}{\pi(\theta \mid x)\mathbb{P}_{\theta,\theta'}(\omega_1, \omega_2)} \cdot \frac{q_{w_1}(\theta', \theta)}{q_{w_1}(\theta, \theta')}, 1 \right\} \right] \right) \lambda_1(\mathrm{d}\omega_1)$$

$$\leq \int_{\Omega_1} \mathbb{P}_\theta(\omega_1) q_{\omega_1}(\theta, \theta') \left( \min \left\{ \frac{\pi(\theta' \mid x)q_{w_1}(\theta', \theta)}{\pi(\theta \mid x)q_{w_1}(\theta, \theta')} \mathbb{E}_{\omega_2 \mid \omega_1, \theta, \theta'} \left[ \frac{\mathbb{P}_{\theta',\theta}(\omega_1, \omega_2)}{\mathbb{P}_{\theta,\theta'}(\omega_1, \omega_2)} \right], 1 \right\} \right) \lambda_1(\mathrm{d}\omega_1)$$

$$= \int_{\Omega_1} \mathbb{P}_\theta(\omega_1) q_{\omega_1}(\theta, \theta') \left( \min \left\{ \frac{\pi(\theta' \mid x)q_{w_1}(\theta', \theta)}{\pi(\theta \mid x)q_{w_1}(\theta, \theta')} \frac{\mathbb{P}_{\theta'}(\omega_1)}{\mathbb{P}_\theta(\omega_1)}, 1 \right\} \right) \lambda_1(\mathrm{d}\omega_1)$$

$$= p_{\mathsf{MwG}}(\theta, \theta')$$

The first inequality follows from $\mathbb{E}[\min\{ah(X), c\}] \leq \min\{a\mathbb{E}[h(X)], c\}$, treating $\omega_2$ as a random variable, and using the fact that

$$\mathbb{E}_{\omega_2|\omega_1,\theta,\theta'}\left[\frac{\mathbb{P}_{\theta',\theta}(\omega_1,\omega_2)}{\mathbb{P}_{\theta,\theta'}(\omega_1,\omega_2)}\right] = \int_{\Omega_2}\frac{\mathbb{P}_{\theta'}(\omega_1)\mathbb{P}_{\theta',\theta}(\omega_2 \mid \omega_1)}{\mathbb{P}_\theta(\omega_1)\mathbb{P}_{\theta,\theta'}(\omega_2 \mid \omega_1)}\mathbb{P}_{\theta,\theta'}(\omega_2 \mid \omega_1)\lambda_2(d\omega_2)$$
$$= \frac{\mathbb{P}_{\theta'}(\omega_1)}{\mathbb{P}_\theta(\omega_1)}.$$

The last equality follows from the calculations in Case 2.

Furthermore

$$p_{\mathsf{MwG}}(\theta, \theta') \leq \min\left\{\int\frac{\pi(\theta' \mid x)}{\pi(\theta \mid x)}q_{w_1}(\theta', \theta)\mathbb{P}_{\theta'}(\omega_1)\lambda_1(d\omega_1), \int q_{\omega_1}(\theta, \theta')\mathbb{P}_\theta(\omega_1)\lambda_1(d\omega_1)\right\}$$
$$= \min\left\{\frac{\pi(\theta' \mid x)}{\pi(\theta \mid x)}q_{\mathsf{ideal}}(\theta', \theta), q_{\mathsf{ideal}}(\theta, \theta')\right\}$$
$$= q_{\mathsf{ideal}}(\theta, \theta')\min\left\{\frac{\pi(\theta' \mid x)}{\pi(\theta \mid x)}\frac{q_{\mathsf{ideal}}(\theta', \theta)}{q_{\mathsf{ideal}}(\theta, \theta')}, 1\right\}$$
$$= p_{\mathsf{ideal}}(\theta, \theta').$$

Here, the inequality uses the fact $\int \min\{f, g\} \leq \min\{\int f, \int g\}$. $\qquad\square$

### B.6. Proof of Theorem 1

*Proof of Theorem 1.* We can write down the transition density as:

$$p_{\mathsf{aux}}(\theta, \theta') = \int_{\Omega_1 \times \Omega_2}\mathbb{P}_\theta(\omega_1)q_{\omega_1}(\theta, \theta')\mathbb{P}_{\theta,\theta'}(\omega_2 \mid \omega_1)r(\theta, \theta'; \omega_1, \omega_2)\lambda_1 \otimes \lambda_2(d\omega_1 d\omega_2)$$
$$= \int_{\Omega_1}\mathbb{P}_\theta(\omega_1)q_{\omega_1}(\theta, \theta')\left(\mathbb{E}_{\omega_2|\omega_1,\theta,\theta'}\left[\min\left\{\frac{\pi(\theta' \mid x)\mathbb{P}_{\theta',\theta}(\omega_1,\omega_2)}{\pi(\theta \mid x)\mathbb{P}_{\theta,\theta'}(\omega_1,\omega_2)}\cdot\frac{q_{w_1}(\theta', \theta)}{q_{w_1}(\theta, \theta')}, 1\right\}\right]\right)\lambda_1(d\omega_1)$$
$$\geq \int_{\Omega_1}\mathbb{P}_\theta(\omega_1)q_{\omega_1}(\theta, \theta')\left(\min\left\{\frac{\pi(\theta' \mid x)q_{w_1}(\theta', \theta)}{\pi(\theta \mid x)q_{w_1}(\theta, \theta')}\frac{\mathbb{P}_{\theta'}(\omega_1)}{\mathbb{P}_\theta(\omega_1)}, 1\right\}\right)$$
$$\times\left(\int_{\Omega_2}\min\left\{\mathbb{P}_{\theta,\theta'}(\omega_2 \mid \omega_1), \mathbb{P}_{\theta',\theta}(\omega_2 \mid \omega_1)\right\}\lambda_2(d\omega_2)\right)\lambda_1(d\omega_1)$$
$$= \int_{\Omega_1}\mathbb{P}_\theta(\omega_1)q_{\omega_1}(\theta, \theta')\left(\min\left\{\frac{\pi(\theta' \mid x)q_{w_1}(\theta', \theta)}{\pi(\theta \mid x)q_{w_1}(\theta, \theta')}\frac{\mathbb{P}_{\theta'}(\omega_1)}{\mathbb{P}_\theta(\omega_1)}, 1\right\}\right)$$
$$\times(1 - d_{\mathsf{TV}}(\theta, \theta', \omega_1))\lambda_1(d\omega_1)$$
$$\geq (1 - d_{\mathsf{TV}}(\theta, \theta'))\int_{\Omega_1}\mathbb{P}_\theta(\omega_1)q_{\omega_1}(\theta, \theta')\left(\min\left\{\frac{\pi(\theta' \mid x)q_{w_1}(\theta', \theta)}{\pi(\theta \mid x)q_{w_1}(\theta, \theta')}\frac{\mathbb{P}_{\theta'}(\omega_1)}{\mathbb{P}_\theta(\omega_1)}, 1\right\}\right)$$
$$= (1 - d_{\mathsf{TV}}(\theta, \theta'))\, p_{\mathsf{MwG}}(\theta, \theta').$$

The first inequality follows from the fact $\min\{ab, cd\} \geq \min\{a, c\}\min\{b, d\}$ for $a, b, c, d \geq 0$. For the second inequality in the statement of Theorem 1, first recall the improved version of Bretagnolle–Huber inequality in Section 8.3 of (Gerchinovitz et al., 2020), which states:

$$1 - d_{\mathsf{TV}}(P, Q) \geq e^{-1/e}\exp\{-d_{\mathsf{KL}}(P||Q)\}.$$

The original Bretagnolle–Huber inequality (Bretagnolle & Huber, 1979; Canonne, 2022) is a slightly weaker version where the constant $e^{-1/e}$ is replaced by $1/2$. Replacing $P, Q$ by $\mathbb{P}_{\theta,\theta'}(\omega_2 \mid \omega_1), \mathbb{P}_{\theta',\theta}(\omega_2 \mid \omega_1)$, and taking the supremum (for $d_{\mathsf{TV}}$) over $\omega_1$, we get:

$$p_{\mathsf{aux}}(\theta, \theta') \geq e^{-1/e}\exp\{-\tilde{d}_{\mathsf{KL}}(\theta, \theta')\}\}p_{\mathsf{MwG}}(\theta, \theta').$$

Meanwhile, reversing the order between $P, Q$ in the Bretagnolle–Huber inequality, we have

$$1 - d_{\mathsf{TV}}(Q, P) \geq e^{-1/e} \exp\{-d_{\mathsf{KL}}(Q||P)\}.$$

This implies

$$p_{\mathsf{aux}}(\theta, \theta') \geq e^{-1/e} \exp\{-\tilde{d}_{\mathsf{KL}}(\theta', \theta)\} p_{\mathsf{MwG}}(\theta, \theta').$$

Finally, multiplying the improved Bretagnolle–Huber inequality for $(P, Q)$ and $(Q, P)$ together and then taking the square root, we obtain:

$$1 - d_{\mathsf{TV}}(P, Q) \geq e^{-1/e} \exp\{-0.5(d_{\mathsf{KL}}(P||Q) + d_{\mathsf{KL}}(Q||P))\},$$

which further shows

$$1 - d_{\mathsf{TV}}(\theta, \theta') \geq e^{-1/e} \exp\{-d_{\mathsf{KL}}(\theta, \theta')\}.$$

This concludes the proof of Theorem 1. $\qquad\square$

## B.7. Spectral Gap

We begin by a concise review of definitions. Consider $P$ as the transition kernel for a Markov chain with stationary distribution $\Pi$. The linear space $L^2(\Pi)$ includes all functions $f$ satisfying $\int f^2(x)\Pi(\mathrm{d}x) < \infty$. Within this space, $L_0^2(\Pi)$ is the subspace of functions $f \in L^2(\Pi)$ satisfying $\int f(x)\Pi(\mathrm{d}x) = 0$. Both $L^2(\Pi)$ and $L_0^2(\Pi)$ are Hilbert spaces with the inner product $\langle f, g \rangle_\Pi := \int (fg)\mathrm{d}\Pi = \mathbb{E}_\Pi[fg]$. The norm of $f \in L^2(\Pi)$ is $\|f\|_\Pi := \sqrt{\langle f, f \rangle_\Pi}$. The kernel $P$ operates as a linear operator on $L_0^2(\Pi)$, acting on $f$ by $(Pf)(x) := \int P(x, \mathrm{d}y)f(y)$. Its operator norm on $L_0^2(\Pi)$ is defined as $\|P\|_{L_0^2(\Pi)} := \sup_{f \in L_0^2(\Pi), \|f\|_\Pi = 1} \|Pf\|_\Pi$. It is known that $\|P\|_{L_0^2(\Pi)} \leq 1$. The spectral gap of $P$, denoted as $\mathsf{Gap}(P)$, is $1 - \|P\|_{L_0^2(\Pi)}$.

The next proposition studies the $\mathsf{Gap}(\mathbb{P}_{\mathsf{aux}})$.

**Theorem 2.** *Let $\Pi = \pi(\theta \mid x)$ be the common stationary distribution of $\mathbb{P}_{\mathsf{MwG}}$ and $\mathbb{P}_{\mathsf{aux}}$. We have*

$$\mathsf{Gap}(\mathbb{P}_{\mathsf{aux}}) \geq (1 - \sup_{\theta, \theta'} d_{\mathsf{TV}}(\theta, \theta'))\mathsf{Gap}(\mathbb{P}_{\mathsf{MwG}}).$$

*Similarly, we have*

$$\mathsf{Gap}(\mathbb{P}_{\mathsf{aux}}) \geq \left(e^{-1/e} \exp\{-\sup_{\theta \neq \theta'} \min\{d_{\mathsf{KL}}(\theta, \theta'), \tilde{d}_{\mathsf{KL}}(\theta, \theta'), \tilde{d}_{\mathsf{KL}}(\theta', \theta)\}\}\right) \mathsf{Gap}(\mathbb{P}_{\mathsf{MwG}}).$$

For an initial distribution $\pi_0$ that is $\beta$-warm with respect to the stationary distribution $\Pi$ (i.e., $\pi_0(A)/\Pi(A) \leq \beta$ for any $A$), a Markov chain $P$ starting from a $\beta$-warm distribution reaches within $\epsilon$ of the stationary distribution in both total-variation and $L^2$ distance after $\mathcal{O}(\log(\beta\epsilon^{-1})/\mathsf{Gap}(P))$ iterations, as shown in Theorem 2.1 of Roberts & Rosenthal (1997). Theorem 2 states that $\mathbb{P}_{\mathsf{aux}}$ is at most $1/(1 - \sup_{\theta, \theta'} d_{\mathsf{TV}}(\theta, \theta'))$ times slower than $\mathbb{P}_{\mathsf{MwG}}$ in terms of iteration count. Conversely, if the per-iteration cost of $\mathbb{P}_{\mathsf{aux}}$ is less than $1/(1 - \sup_{\theta, \theta'} d_{\mathsf{TV}}(\theta, \theta'))$ times the cost of $\mathbb{P}_{\mathsf{MwG}}$, then $\mathbb{P}_{\mathsf{aux}}$ is provably more efficient than $\mathbb{P}_{\mathsf{MwG}}$.

**Bounding $\mathsf{Gap}(\mathbb{P}_{\mathsf{MwG}})$:** Given the relationship between $\mathsf{Gap}(\mathbb{P}_{\mathsf{aux}})$ and $\mathsf{Gap}(\mathbb{P}_{\mathsf{MwG}})$ in Theorem 2, the next step is to bound $\mathsf{Gap}(\mathbb{P}_{\mathsf{MwG}})$. We now discuss recent findings from (Qin et al., 2025). In each iteration, the chain $\mathbb{P}_{\mathsf{MwG}}$ first draws $\omega_1 \sim \mathbb{P}_\theta(\cdot)$, followed by a Metropolis–Hastings algorithm to sample from $\pi(\theta \mid \omega_1, x) \propto \pi(\theta \mid x)\mathbb{P}_\theta(\omega_1)$ using the proposal $q_{\omega_1}(\theta, \cdot)$. We use $\mathbb{P}_{\mathsf{MHG}, \omega_1}$ to denote the transition kernel of the second step. We also use $\mathbb{P}_{\mathsf{Gibbs}}$ to denote the "$\theta$-chain" of the standard Gibbs sampler targeting $\pi(\theta \mid x)\mathbb{P}_\theta(\omega_1)$. This means we first draw $\omega_1 \sim \mathbb{P}_\theta(\cdot)$, and then draw $\theta$ directly from $\pi(\theta \mid \omega_1, x)$.

**Proposition 6** (Corollary 14 and Theorem 15 of (Qin et al., 2025))**.** *We have:*

1. $\mathsf{Gap}(\mathbb{P}_{\mathsf{MwG}}) \geq \mathsf{Gap}(\mathbb{P}_{\mathsf{Gibbs}}) \times \inf_{\omega_1} \mathsf{Gap}(\mathbb{P}_{\mathsf{MHG}, \omega_1})$

2. *Suppose there is a function $\gamma_1 : \Omega_1 \to [0, 1]$ satisfying $\|\mathbb{P}_{\mathsf{MHG},\omega_1}\|_{L_0^2(\pi(\cdot|\omega_1,x))} \leq \gamma_1(\omega_1)$ for every $\omega_1$. Meanwhile $\int \gamma_1(\omega_1)^t \mathbb{P}_\theta(\mathrm{d}\omega_1) \leq \alpha_t$ for every $\theta$ and some even $t$. Then $\mathsf{Gap}(\mathbb{P}_{\mathsf{MwG}}) \geq t^{-1} (\mathsf{Gap}(\mathbb{P}_{\mathsf{Gibbs}}) - \alpha_t)$*

The first result is known in (Andrieu et al., 2018b). The second result generalizes the results of (Łatuszyński & Rudolf, 2024). They have been applied in analyzing proximal samplers and hybrid slice samplers. See Section 5 of (Qin et al., 2025) for details.

It is useful to recall the definition of the Dirichlet form. The Dirichlet form of a function $f$ with respect to Markov transition kernel $P$ is $\mathcal{E}(P, f) := \|(I - P)f, f\|_\Pi = 0.5 \int \int (f(x) - f(y))^2 \Pi(\mathrm{d}x)P(x, \mathrm{d}y)$. When $P$ is reversible, then $\mathsf{Gap}(P) = \inf_{f \in L_0^2(\Pi), f \neq 0} \mathcal{E}(P, f)/\|f\|_\Pi^2$.

*Proof of Theorem 2.* It follows from Theorem 1 that for any $f$, the Dirichlet form satisfies

$$\mathcal{E}(\mathbb{P}_{\mathsf{aux}}, f) \geq \int \int (1 - d_{\mathsf{TV}}(\theta, \theta')) (f(\theta) - f(\theta'))^2 \mathbb{P}_{\mathsf{MwG}}(\theta, \mathrm{d}\theta')\pi(\mathrm{d}\theta \mid x)$$

$$\geq \left( 1 - \sup_{\theta, \theta'} d_{\mathsf{TV}}(\theta, \theta') \right) \mathcal{E}(\mathbb{P}_{\mathsf{MwG}}, f).$$

Dividing both sides by $\|f\|_\Pi^2$ and then taking the infimum over $f \in L_0^2(\pi(\theta \mid x))$, we have

$$\mathsf{Gap}(\mathbb{P}_{\mathsf{aux}}) \geq (1 - \sup_{\theta, \theta'} d_{\mathsf{TV}}(\theta, \theta'))\mathsf{Gap}(\mathbb{P}_{\mathsf{MwG}}).$$

Similarly, we have

$$\mathcal{E}(\mathbb{P}_{\mathsf{aux}}, f) \geq e^{-1/e} \exp\{- \sup_{\theta \neq \theta'} \min\{d_{\mathsf{KL}}(\theta, \theta'), \tilde{d}_{\mathsf{KL}}(\theta, \theta'), \tilde{d}_{\mathsf{KL}}(\theta', \theta)\}\}\mathcal{E}(\mathbb{P}_{\mathsf{MwG}}, f).$$

Therefore $\mathsf{Gap}(\mathbb{P}_{\mathsf{aux}}) \geq \left( e^{-1/e} \exp\{-\sup_{\theta \neq \theta'} \min\{d_{\mathsf{KL}}(\theta, \theta'), \tilde{d}_{\mathsf{KL}}(\theta, \theta'), \tilde{d}_{\mathsf{KL}}(\theta', \theta)\}\} \right) \mathsf{Gap}(\mathbb{P}_{\mathsf{MwG}})$. $\qquad\square$

### B.8. Proof of Proposition 3

*Proof of Proposition 3.* Our first result is derived directly by recognizing that $\mathbb{P}_{\theta,\theta'}$ within our general framework corresponds to $p_{\theta'}$ in the exchange algorithm. The second result follows from the observation:

$$1 - d_{\mathsf{TV}}(p_\theta, p_{\theta'}) = \mathbb{E}_{p_{\theta'}} \left[ \min\{1, \frac{p_\theta}{p_{\theta'}}\} \right] \geq \delta \mathbb{P}_{p_{\theta'}} (A_{\theta,\theta'}(\delta)) = \epsilon \delta.$$

$\qquad\square$

### B.9. Proof of Proposition 4

*Proof of Proposition 4.* Let

$$\widetilde{\lambda}_i(\theta) := \frac{\lambda M_i}{L} + \phi_i(\theta; x).$$

In PoissonMH,

$$\mathbb{P}_{\theta,\theta'} = \bigotimes_{i=1}^N \mathsf{Poi}\left(\widetilde{\lambda}_i(\theta)\right).$$

Since PoissonMH does not use an auxiliary variable in the proposal step, we have $\mathbb{P}_{\mathsf{MwG}} = \mathbb{P}_{\mathsf{ideal}}$.

We first derive the new comparison bound from the symmetrized KL part of Theorem 2. For product Poisson laws,

$$
d_{\mathsf{KL}}(\mathbb{P}_{\theta,\theta'} \| \mathbb{P}_{\theta',\theta}) + d_{\mathsf{KL}}(\mathbb{P}_{\theta',\theta} \| \mathbb{P}_{\theta,\theta'})
$$

$$
= \sum_{i=1}^{N} \left[ \widetilde{\lambda}_i(\theta) \log \frac{\widetilde{\lambda}_i(\theta)}{\widetilde{\lambda}_i(\theta')} + \widetilde{\lambda}_i(\theta') - \widetilde{\lambda}_i(\theta) \right]
$$

$$
+ \sum_{i=1}^{N} \left[ \widetilde{\lambda}_i(\theta') \log \frac{\widetilde{\lambda}_i(\theta')}{\widetilde{\lambda}_i(\theta)} + \widetilde{\lambda}_i(\theta) - \widetilde{\lambda}_i(\theta') \right]
$$

$$
= \sum_{i=1}^{N} \left( \widetilde{\lambda}_i(\theta) - \widetilde{\lambda}_i(\theta') \right) \log \frac{\widetilde{\lambda}_i(\theta)}{\widetilde{\lambda}_i(\theta')}.
$$

For every $i$,

$$
\frac{\lambda M_i}{L} \leq \widetilde{\lambda}_i(\theta) \leq \frac{\lambda M_i}{L} + M_i = \frac{(\lambda + L) M_i}{L},
$$

and hence

$$
\left| \log \frac{\widetilde{\lambda}_i(\theta)}{\widetilde{\lambda}_i(\theta')} \right| \leq \log \left( 1 + \frac{L}{\lambda} \right).
$$

Also,

$$
\left| \widetilde{\lambda}_i(\theta) - \widetilde{\lambda}_i(\theta') \right| = |\phi_i(\theta; x) - \phi_i(\theta'; x)| \leq M_i.
$$

Therefore,

$$
d_{\mathsf{KL}}(\mathbb{P}_{\theta,\theta'} \| \mathbb{P}_{\theta',\theta}) + d_{\mathsf{KL}}(\mathbb{P}_{\theta',\theta} \| \mathbb{P}_{\theta,\theta'})
$$

$$
\leq \log \left( 1 + \frac{L}{\lambda} \right) \sum_{i=1}^{N} M_i = L \log \left( 1 + \frac{L}{\lambda} \right).
$$

Thus the symmetrized KL quantity in Theorem 2 satisfies

$$
d_{\mathsf{KL}}(\theta, \theta') \leq \frac{L}{2} \log \left( 1 + \frac{L}{\lambda} \right),
$$

uniformly over $\theta, \theta'$. Applying Theorem 2 gives

$$
\mathsf{Gap}(\mathbb{P}_{\mathsf{aux}}) \geq e^{-1/e} \left( 1 + \frac{L}{\lambda} \right)^{-L/2} \mathsf{Gap}(\mathbb{P}_{\mathsf{ideal}}).
$$

Using $\log(1 + x) \leq x$, we further obtain

$$
\mathsf{Gap}(\mathbb{P}_{\mathsf{aux}}) \geq e^{-1/e} \exp \left\{ -\frac{L^2}{2\lambda} \right\} \mathsf{Gap}(\mathbb{P}_{\mathsf{ideal}}).
$$

On the other hand, the standard PoissonMH transition-density comparison gives

$$
\mathsf{Gap}(\mathbb{P}_{\mathsf{aux}}) \geq \frac{1}{2} \exp \left\{ -\frac{L^2}{\lambda + L} \right\} \mathsf{Gap}(\mathbb{P}_{\mathsf{ideal}}).
$$

Combining the two bounds yields

$$
\mathsf{Gap}(\mathbb{P}_{\mathsf{aux}}) \geq \max \left\{ \frac{1}{2} \exp \left( -\frac{L^2}{\lambda + L} \right), \ e^{-1/e} \exp \left( -\frac{L^2}{2\lambda} \right) \right\} \mathsf{Gap}(\mathbb{P}_{\mathsf{ideal}}).
$$

$\square$

**B.10. Proof of Proposition 5**

*Proof of Proposition 5.* Let $\tilde{\phi}_i(\theta, \theta') := \lambda c_i / C + \phi_i(\theta, \theta'; x)$. We have $\mathbb{P}_{\theta, \theta'} = \otimes_{i=1}^N \text{Poi}(\tilde{\phi}_i(\theta, \theta'))$ for TunaMH. We can calculate the symmetrized KL divergence

$$
\begin{aligned}
&d_{\mathsf{KL}}(\mathbb{P}_{\theta,\theta'} || \mathbb{P}_{\theta',\theta}) + d_{\mathsf{KL}}(\mathbb{P}_{\theta',\theta} || \mathbb{P}_{\theta,\theta'}) \\
&= \sum_{i=1}^N \left( \tilde{\phi}_i(\theta, \theta') \log \frac{\tilde{\phi}_i(\theta, \theta')}{\tilde{\phi}_i(\theta', \theta)} + \cancel{\tilde{\phi}_i(\theta', \theta)} - \cancel{\tilde{\phi}_i(\theta, \theta')} \right) + \sum_{i=1}^N \left( \tilde{\phi}_i(\theta', \theta) \log \frac{\tilde{\phi}_i(\theta', \theta)}{\tilde{\phi}_i(\theta, \theta')} + \cancel{\tilde{\phi}_i(\theta, \theta')} - \cancel{\tilde{\phi}_i(\theta', \theta)} \right) \\
&= \sum_{i=1}^N \left( \log \left( \tilde{\phi}_i(\theta, \theta') / \tilde{\phi}_i(\theta', \theta) \right) \right) \left( \tilde{\phi}_i(\theta, \theta') - \tilde{\phi}_i(\theta', \theta) \right) \\
&\leq \sum_{i=1}^N c_i M(\theta, \theta') \log \left( \frac{\lambda c_i / C + c_i M}{\lambda c_i / C} \right) \\
&\leq \sum_{i=1}^N c_i M(\theta, \theta') \frac{C M(\theta, \theta')}{\lambda} = \frac{C^2 M^2(\theta, \theta')}{\chi C^2 M^2(\theta, \theta')} = \frac{1}{\chi}.
\end{aligned}
$$

Therefore, applying Theorem 2, we get:

$$
\mathsf{Gap}(\mathbb{P}_{\mathsf{aux}}) \geq e^{-1/e} \exp\{-\frac{1}{2\chi}\} \mathsf{Gap}(\mathbb{P}_{\mathsf{ideal}}).
$$

$\square$

**B.11. Proof of Auxiliary Results**

Here we prove several auxiliary results. Even though many of these results are well-known, we include proofs here for clarity and completeness.

**Proposition 7.** *For univariate Poisson random variables with parameter $\lambda_1, \lambda_2$, their KL divergence equals* $d_{\mathsf{KL}}(\text{Poi}(\lambda_1) || \text{Poi}(\lambda_2)) = \lambda_1 \log(\lambda_1 / \lambda_2) + \lambda_2 - \lambda_1$.

*Proof of Proposition 7.* Let

$$
P_\lambda(n) := \exp\{-\lambda\} \frac{\lambda^n}{n!}
$$

be the probability mass function of a $\text{Poi}(\lambda)$ random variable evaluating at $n \in \mathbb{N}$. By definition, we have:

$$
d_{\mathsf{KL}}(\text{Poi}(\lambda_1) || \text{Poi}(\lambda_2)) = \mathbb{E}_{X \sim \text{Poi}(\lambda_1)} \left[ \log \left( \frac{P_{\lambda_1}(X)}{P_{\lambda_2}(X)} \right) \right].
$$

Therefore,

$$
\begin{aligned}
d_{\mathsf{KL}}(\text{Poi}(\lambda_1) || \text{Poi}(\lambda_2)) &= \mathbb{E}_{X \sim \text{Poi}(\lambda_1)} \left[ -\lambda_1 + \lambda_2 + X \log \left( \frac{\lambda_1}{\lambda_2} \right) \right] \\
&= \lambda_2 - \lambda_1 + \log \left( \frac{\lambda_1}{\lambda_2} \right) \mathbb{E}_{X \sim \text{Poi}(\lambda_1)}[X] \\
&= \lambda_1 \log \left( \frac{\lambda_1}{\lambda_2} \right) + \lambda_2 - \lambda_1.
\end{aligned}
$$

$\square$

**Proposition 8.** *For any $\lambda_1, \lambda_2$ satisfying $0 \leq m \leq \lambda_1 \leq \lambda_2 \leq M$, we have*

$$
d_{\mathsf{KL}}(\text{Poi}(\lambda_1) || \text{Poi}(\lambda_2)) \leq d_{\mathsf{KL}}(\text{Poi}(m) || \text{Poi}(M)).
$$

*Proof of Proposition 8.* The proposition is equivalent to proving that the function $g(\lambda_1, \lambda_2) := \lambda_1 \log(\lambda_1/\lambda_2) + \lambda_2 - \lambda_1$, constrained within the region $m \leq \lambda_1 \leq \lambda_2 \leq M$, is maximized at $\lambda_1 = m$ and $\lambda_2 = M$. We first fix $\lambda_1$, and take derivative with respect to $\lambda_2$:

$$\partial_{\lambda_2} g(\lambda_1, \lambda_2) = -\frac{\lambda_1}{\lambda_2} + 1.$$

The derivative is non-negative as $\lambda_2 \geq \lambda_1$. Therefore, $g(\lambda_1, \lambda_2) \leq g(\lambda_1, M)$. Then we arbitrarily fix $\lambda_2$, and take derivative with respect to $\lambda_1$:

$$\partial_{\lambda_1} g(\lambda_1, \lambda_2) = \log(\lambda_1) + 1 - \log(\lambda_2) - 1 = \log(\lambda_1) - \log(\lambda_2) \leq 0.$$

Therefore, $g(\lambda_1, \lambda_2) \leq g(m, \lambda_2)$. Putting these two inequalities together, we know:

$$d_{\mathsf{KL}}(\mathsf{Poi}(\lambda_1)||\mathsf{Poi}(\lambda_2)) = g(\lambda_1, \lambda_2) \leq g(m, M) = d_{\mathsf{KL}}(\mathsf{Poi}(m)||\mathsf{Poi}(M)).$$

$\square$

**Proposition 9.** *Let $P_i, Q_i$ be two probability measures on the same measurable space $(\Omega_i, \mathcal{F}_i)$ for $i \in \{1, 2, \ldots, M\}$. Then we have*

$$d_{\mathsf{KL}}(\otimes_{i=1}^{M} P_i || \otimes_i Q_i) = \sum_{i=1}^{M} d_{\mathsf{KL}}(P_i||Q_i)$$

*and*

$$1 - d_{\mathsf{TV}}(\otimes_{i=1}^{M} P_i, \otimes_i Q_i) \geq \prod_{i=1}^{M}(1 - d_{\mathsf{TV}}(P_i, Q_i)).$$

*Proof.* Let $P := \otimes_{i=1}^{M} P_i$ and $Q := \otimes_{i=1}^{M} Q_i$, and $X = (X_1, \ldots, X_M) \sim P$. Then

$$d_{\mathsf{KL}}(P||Q) = \mathbb{E}_{X \sim P}\left[\log\left(\frac{P(X)}{Q(X)}\right)\right]$$

$$= \mathbb{E}_{X \sim P}\left[\sum_{i=1}^{M} \log\left(\frac{P(X_i)}{Q(X_i)}\right)\right]$$

$$= \sum_{i=1}^{M} \mathbb{E}_{X_i \sim P_i}\left[\log\left(\frac{P(X_i)}{Q(X_i)}\right)\right]$$

$$= \sum_{i=1}^{M} d_{\mathsf{KL}}(P_i||Q_i),$$

which proves the first equality.

For the second equality, recall the fact $d_{\mathsf{TV}}(\mu, \nu) = 1 - \sup_{(X,Y) \in \Gamma(\mu, \nu)} \mathbb{P}(X = Y)$. Here, $\Gamma(\mu, \nu)$ denotes all the couplings of $\mu$ and $\nu$ (that is, all the joint distributions with the marginal distributions $\mu$ and $\nu$). Now fix any $\epsilon > 0$, for each $i \in \{1, \ldots, M\}$, choose $\Pi_i \in \Gamma(P_i, Q_i)$ such that

$$\mathbb{P}_{(X_i, Y_i) \sim \Pi_i}(X = Y) \geq 1 - d_{\mathsf{TV}}(P_i, Q_i) - \epsilon.$$

Consider the $2M$-dimensional vector $(X_1, Y_1, X_2, Y_2, \ldots, X_M, Y_M) \sim \prod_{i=1}^{M} \Pi_i$, then the joint distribution of $X = (X_1, \ldots, X_M)$ and $Y = (Y_1, \ldots, Y_M)$ belongs to the $\Gamma(P, Q)$. Hence

$$1 - d_{\mathsf{TV}}(P, Q) \geq \mathbb{P}(X = Y) = \prod_{i=1}^{n} \mathbb{P}(X_i = Y_i) \geq \prod_{i=1}^{n}(1 - d_{\mathsf{TV}}(P_i, Q_i) - \epsilon).$$

Since $\epsilon$ is an arbitrary positive constant, we can let $\epsilon \to 0$ and get

$$1 - d_{\mathsf{TV}}(\otimes_{i=1}^{M} P_i, \otimes_i Q_i) \geq \prod_{i=1}^{M}(1 - d_{\mathsf{TV}}(P_i, Q_i)).$$

$\square$

### B.12. Additional Properties

**Proposition 10.** *Let $a : [0, \infty) \to [0, 1]$ be any fixed function satisfying $a(t) = ta(1/t)$. Let $\tilde{\mathbb{P}}_{\text{aux}}(\cdot, \cdot)$ denote the transition kernel of the Markov chain as defined in Algorithm 1, with the exception that Step 7 is modified to: "with probability $a(r)$, set $\theta_{t+1} \leftarrow \theta'$; otherwise, set $\theta_{t+1} \leftarrow \theta_t$". Then $\tilde{\mathbb{P}}_{\text{aux}}(\cdot, \cdot)$ remains reversible with respect to $\pi(\theta \mid x)$.*

*Proof of Proposition 10.* Let $\tilde{p}_{\text{aux}}$ denote the density part of $\tilde{P}_{\text{aux}}$. We have the following expression of $\tilde{p}_{\text{aux}}(\theta, \theta')$:

$$
\begin{aligned}
&\tilde{p}_{\text{aux}}(\theta, \theta') \\
&= \int_{\Omega_1} \mathbb{P}_\theta(\omega_1) q_{\omega_1}(\theta, \theta') \int_{\Omega_2} \left( a\left( \frac{\Pi(\theta')\mathbb{P}_{\theta',\theta}(\omega_1, \omega_2)}{\Pi(\theta)\mathbb{P}_{\theta,\theta'}(\omega_1, \omega_2)} \cdot \frac{q_{w_1}(\theta', \theta)}{q_{w_1}(\theta, \theta')} \right) \mathbb{P}_{\theta,\theta'}(\omega_2 \mid \omega_1) \lambda_2(\mathrm{d}\omega_2) \right) \lambda_1(\mathrm{d}\omega_1) \\
&= \frac{\Pi(\theta')}{\Pi(\theta)} \int_{\Omega_1} \int_{\Omega_2} \mathbb{P}_{\theta',\theta}(\omega_1, \omega_2) q_{\omega_1}(\theta', \theta) a\left( \frac{\Pi(\theta)\mathbb{P}_{\theta,\theta'}(\omega_1, \omega_2)}{\Pi(\theta')\mathbb{P}_{\theta',\theta}(\omega_1, \omega_2)} \cdot \frac{q_{w_1}(\theta, \theta')}{q_{w_1}(\theta', \theta)} \right) \lambda_2(\mathrm{d}\omega_2) \lambda_1(\mathrm{d}\omega_1) \\
&= \frac{\Pi(\theta')}{\Pi(\theta)} \tilde{p}_{\text{aux}}(\theta', \theta),
\end{aligned}
$$

where the second equality uses $a(r) = ra(1/r)$. Therefore we have

$$
\Pi(\theta)\tilde{p}_{\text{aux}}(\theta, \theta') = \Pi(\theta')\tilde{p}_{\text{aux}}(\theta', \theta),
$$

as desired. $\qquad\square$

## C. Implementation Details of Poisson Minibatching

We provide the implementation details for sampling the minibatch data in Algorithm 8 (for Algorithm 5, 2), and Algorithm 9 (for Algorithm 6, 3). The goal is to demonstrate that this minibatch sampling step can be implemented at a much lower cost than its naïve implementation (and also lower than full-batch MCMC). All the material here is not new, as this subsampling trick has been extensively discussed in (Zhang & De Sa, 2019; Zhang et al., 2020). Since the same subsampling method is used in our new algorithm (Algorithm 2, 3), we include their details here for concreteness.

---

**Algorithm 8** Minibatch Sampling in PoissonMH and Locally Balanced PoissonMH

---

▷ One time pre-computation
- set $\Lambda_i := \frac{\lambda M_i}{L} + M_i$ and $\Lambda := \sum_{i=1}^N \Lambda_i$,
- construct an alias table for the categorical distribution on $\{1, 2, \cdots, N\}$: `Categorical`$(p_1, p_2, \cdots, p_N)$, where $p_i = \Lambda_i / \Lambda$

▷ Poisson minibatch sampling (Step 5 of Algorithm 5, Step 3 of Algorithm 2)
**Require:** Current position $\theta$
  Initialize $s_i = 0$ for $i \in \{1, 2, \ldots, N\}$
  sample $B \sim \mathsf{Poi}(\Lambda)$
  **for** $b = 1$ to $B$ **do**
    sample $i_b \sim$ `Categorical`$(p_1, p_2, \cdots, p_N)$
    compute $\phi_{i_b}(\theta; x)$
    with probability $\frac{\frac{\lambda M_{i_b}}{L} + \phi_{i_b}(\theta; x)}{\frac{\lambda M_{i_b}}{L} + M_{i_b}}$, set $s_{i_b} \leftarrow s_{i_b} + 1$
  **end for**

---

Both algorithms require pre-computation that only needs to be performed once prior to the MCMC iterations. This computation constructs an 'alias table' for sampling the categorical distribution on $\{1, 2, \cdots, N\}$. With this table, drawing a sample from this categorical distribution requires only $\mathcal{O}(1)$ time. See part A and part B in Section III of (Vose, 1991) for details.

The computational cost of generating Poisson variables at each MCMC step is then $\mathcal{O}(B)$. The expected computational cost will be:

---

**Algorithm 9** Minibatch Sampling in TunaMH and Tuna–SGLD

▷ One time pre-computation before MCMC iterations

- construct an alias table for the categorical distribution on $\{1, 2, \cdots, N\}$: `Categorical`$(p_1, p_2, \cdots, p_N)$, where $p_i = c_i/C$

▷ Poisson minibatch sampling (Step 6 of Algorithm 6, Step 5 of Algorithm 3)

**Require:** Current position $\theta$ and proposed state $\theta'$

Initialize $s_i = 0$ for $i \in \{1, 2, \ldots, N\}$

sample $B \sim \mathsf{Poi}(\lambda + CM(\theta, \theta'))$

**for** $b = 1$ to $B$ **do**

    sample $i_b \sim$ `Categorical`$(p_1, p_2, \cdots, p_N)$

    compute $\phi_{i_b}(\theta, \theta'; x)$

    with probability $\frac{\lambda c_{i_b} + C\phi_{i_b}(\theta, \theta'; x)}{\lambda c_{i_b} + Cc_{i_b}M(\theta, \theta')}$, set $s_{i_b} \leftarrow s_{i_b} + 1$

**end for**

---

$$\mathbb{E}[B] = \begin{cases} \lambda + L & \text{for PoissonMH} \\ \lambda + CM(\theta, \theta') & \text{for TunaMH} \end{cases}$$

Here $\lambda$ is our tuning parameter, and $L$ and $C$ are much smaller than $N$ in the examples considered here. Therefore the per-iteration cost is of these minibatch MCMC algorithm is much smaller than standard MCMC.

Next, we verify that Algorithm 8 indeed generates independent Poisson variables required by Step 5 of PoissonMH (Algorithm 5). The validity proof of Algorithm 9 is similar. For PoissonMH, we can view the generating process of $\{s_i\}_{i=1}^N$ as the following urn model: 1. Generates $\mathsf{Poi}(\Lambda)$ balls; 2. Throw each ball to $N$ urns independently, with probability $p_1, p_2, \ldots, p_N$ for the first, second, ..., last urn. 3. For every ball in urn $i$ (if any), keep it with probability $\frac{\frac{\lambda M_{i_b}}{L} + \phi_{i_b}(\theta_t; x)}{\frac{\lambda M_{i_b}}{L} + M_{i_b}}$ and discard it otherwise. We can verify $s_1, s_2, \ldots, s_N$ are the number of balls in urn $1, 2, \ldots, N$ respectively.

On the other hand, step 1 - 3 are precisely the Poisson thinning process. Standard Poisson calculation shows $s_1, s_2, \ldots, s_N$ are independent Poisson random variables, and the Poisson parameter of $s_i$ equals $\Lambda p_i \times \frac{\frac{\lambda M_i}{L} + \phi_i(\theta_t; x)}{\frac{\lambda M_i}{L} + M_i}$. In our case, $p_i = \Lambda_i/\Lambda$, and $\Lambda_i = \lambda M_i/L + M_i$. Therefore $s_i \sim \mathsf{Poi}(\lambda M_i/L + \phi_i(\theta, x))$.

As a result, Algorithm 8 gives the required Poisson variables by PoissonMH. For TunaMH, the proof is similar and is omitted here.

## D. Implementation Details of Locally Balanced PoissonMH

According to (Livingstone & Zanella, 2022), there are two different implementation versions of the locally balanced proposal on $\mathbb{R}^d$. The first approach is to treat each dimension of the parameter space separately. We define the proposal for dimension $j$ as:

$$q^{(g)}_{\omega_1, j}(\theta, \theta'_j) := Z^{-1}(\theta, \omega_1)g\left(e^{\partial_{\theta_j} \log(\pi(\theta|x)\mathbb{P}_\theta(\omega_1))(\theta'_j - \theta_j)}\right)\mu_j(\theta'_j - \theta_j),$$

where $\mu_j$ is a univariate symmetric density. The proposal on $\mathbb{R}^d$ is then defined as the product of proposals on each dimension:

$$q^{(g)}_{\omega_1}(\theta, \theta') = \prod_{j=1}^d q^{(g)}_{\omega_1, j}(\theta, \theta'_j)$$

The second approach is to derive a multivariate version of the locally balanced proposal. Proposition 3.3 and the experiments in (Livingstone & Zanella, 2022) show that the first approach is more efficient. As a result, we develop the auxiliary version of the first approach for both Poisson–Barker and Poisson–MALA.

Now, we provide implementation details of Poisson–MALA and Poisson–Barker. First, by taking $g(t) = \sqrt{t}$ and $\mu_j \sim \mathbb{N}(0, \sigma^2)$ for every $j$, we derive the Poisson–MALA proposal as:

$$q_{\omega_1}^M(\theta, \theta') = \prod_{j=1}^{d} q_{\omega_1, j}^M(\theta, \theta'_j).$$

The $j$-th term in the product can be further represented as

$$q_{\omega_1, j}^M(\theta, \theta'_j) \propto \exp\left\{\frac{1}{2}\partial_{\theta_j} \log\left(\pi(\theta \mid x)\mathbb{P}_\theta(\omega_1)\right)(\theta' - \theta_j)\right\} \exp\left\{-\frac{1}{2\sigma^2}(\theta' - \theta_j)^2\right\}$$

$$\propto \exp\left\{-\frac{1}{2\sigma^2}\left(\theta' - \theta_j - \frac{\sigma^2}{2}\partial_{\theta_j} \log\left(\pi(\theta \mid x)\mathbb{P}_\theta(\omega_1)\right)\right)^2\right\}$$

which is Gaussian centred at $\theta_j + \frac{\sigma^2}{2}\partial_{\theta_j} \log\left(\pi(\theta \mid x)\mathbb{P}_\theta(\omega_1)\right)$ with standard deviation $\sigma$. This proposal is similar to MALA except that we substitute the gradient of $\log \pi(\theta \mid x)$ by the gradient of $\log \pi(\theta \mid x)\mathbb{P}_\theta(\omega_1)$, enabling efficient implementation. Since all the dimensions are independent, our Poisson–MALA proposal is a multivariate Gaussian with mean $\theta + 0.5\sigma^2\nabla_\theta \log\left(\pi(\theta \mid x)\mathbb{P}_\theta(\omega_1)\right)$ and covariance matrix $\sigma^2\mathbb{I}_d$.

Next, when taking $g(t) = t/(1 + t)$, we can similarly write our Poisson–Barker proposal as:

$$q_{\omega_1}^B(\theta, \theta') = \prod_{j=1}^{d} q_{\omega_1, j}^B(\theta, \theta'_j).$$

The $j$-th term in the product can be further represented as

$$q_{\omega_1, j}^B(\theta, \theta'_j) = Z^{-1}(\theta, \omega_1)\frac{\mu_j(\theta'_j - \theta)}{1 + e^{-\partial_{\theta_j} \log(\pi(\theta|x)\mathbb{P}_\theta(\omega_1))(\theta'_j - \theta_j)}}.$$

Now we claim $Z(\theta, \omega_1)$ is a constant 0.5. To verify our claim, we first notice that $g(t) = t/(1 + t) = 1 - g(t^{-1})$. Let $C_{\theta, \omega_1} := \partial_{\theta_j} \log\left(\pi(\theta \mid x)\mathbb{P}_\theta(\omega_1)\right)$, we can integrate:

$$\begin{aligned} Z(\theta, \omega_1) &= \int_{\mathbb{R}} g(\exp\{C_{\theta, \omega_1}(\theta'_j - \theta_j)\})\mu_j(\theta'_j - \theta_j)\mathrm{d}\theta'_j \\ &= \int_{\mathbb{R}} g(\exp\{C_{\theta, \omega_1} t\})\mu_j(t)\mathrm{d}t \\ &= \int_{t>0} g(\exp\{C_{\theta, \omega_1} t\})\mu_j(t)\mathrm{d}t + \int_{t<0} g(\exp\{C_{\theta, \omega_1} t\})\mu_j(t)\mathrm{d}t \\ &= \int_{t>0} g(\exp\{C_{\theta, \omega_1} t\})\mu_j(t)\mathrm{d}t + \int_{t>0} g(\exp\{-C_{\theta, \omega_1} t\})\mu_j(t)\mathrm{d}t \quad \text{since } \mu_j(t) = \mu_j(-t) \\ &= \int_{t>0} \mu_j(t)\mathrm{d}t = 0.5. \end{aligned}$$

The same calculation is used in (Livingstone & Zanella, 2022). Therefore our proposal has an explicit form

$$q_{\omega_1, j}^B(\theta, \theta'_j) = 2\frac{\mu_j(\theta'_j - \theta_j)}{1 + e^{-\partial_{\theta_j} \log(\pi(\theta|x)\mathbb{P}_\theta(\omega_1))(\theta'_j - \theta_j)}}.$$

Sampling from this proposal distribution can be done perfectly by the following simple procedure: given current $\theta_j$, one first propose $z \sim \mu_j$, then move to $\theta_j + z$ with probability $g(e^{\partial_{\theta_j} \log(\pi(\theta|x)\mathbb{P}_\theta(\omega_1))z})$, and to $\theta_j - z$ with probability $1 - g(e^{\partial_{\theta_j} \log(\pi(\theta|x)\mathbb{P}_\theta(\omega_1))z})$. This procedure samples exactly from $q_{\omega_1, j}^B(\theta, \theta'_j)$, see also Proposition 3.1 of (Livingstone & Zanella, 2022) for details. Sampling from the $d$-dimensional proposal can be done by implementing the above strategy for each dimensional independently.

Both algorithms require evaluating $\nabla \log\left(\pi(\theta \mid x)\mathbb{P}_\theta(\omega_1)\right)$, and we verify that this only requires evaluation on the same minibatch as PoissonMH. As derived in Section 3.2.1,

$$\pi(\theta \mid x)\mathbb{P}_\theta(\omega_1) = \frac{1}{Z(x)}\exp(-\lambda)\prod_{i:s_i>0}\frac{\left(\lambda M_i L^{-1} + \phi_i(\theta;x)\right)^{s_i}}{s_i!}.$$

Then, we take the logarithm:

$$\log\pi(\theta \mid x)\mathbb{P}_\theta(\omega_1) = -\log Z(x) - \lambda - \sum_{i:s_i>0}\log(s_i!) + \sum_{i:s_i>0}s_i\log\left\{\lambda M_i L^{-1} + \phi_i(\theta;x)\right\}.$$

The gradient is:

$$\nabla_\theta\log\pi(\theta \mid x)\mathbb{P}_\theta(\omega_1) = \sum_{i:s_i>0}s_i\frac{\nabla_\theta\phi_i(\theta;x)}{\lambda M_i L^{-1} + \phi_i(\theta;x)}$$

As a result, we only need to evaluate the gradient on the minibatch data.

# E. Additional Comment on Algorithm 5 and 6

A close examination of Algorithm 5 and 6 shows that the key ingredient is the 'Poisson estimator', which provides an unbiased estimate of the exponential of an expectation. In the case of tall dataset, the quantity of interest is the target ratio which can be represented as $\exp\left\{\sum_{i=1}^{N}(\phi_i(\theta';x) - \phi_i(\theta;x))\right\}$ in Algorithm 5 (or $\exp\left\{(-\sum_{i=1}^{N}U_i(\theta';x) + U_i(\theta;x))\right\}$ in Algorithm 6). This ratio can be evaluated precisely, albeit at a high cost. Alternatively, we can express the ratio as $\exp\left(\mathbb{E}[f(I)]\right)$, where $f(i) := N(\phi_i(\theta';x) - \phi_i(\theta;x))$ and $I \sim \mathsf{Unif}\{1, 2, \ldots, N\}$. This formulation aligns the problem with the Poisson estimator's framework. An additional subtlety is the estimator has to be non-negative, as it will be used to determine the acceptance probability (Step 7 in Algorithm 5, and Step 6 in Algorithm 3). Consequently, all the technical assumptions and design elements in both Algorithm 5 and 6 are intended to ensure that the non-negativity of the Poisson estimator. Other applications of the Poisson estimator can be found in Fearnhead et al. (2010); Wagner (1987); Papaspiliopoulos (2011) and the references therein. This insight can be interpreted either positively or negatively: it implies that there is potential for developing new algorithms using similar ideas under weaker assumptions (such as non-i.i.d. data or less stringent technical conditions). However, eliminating all technical assumptions seems unlikely due to the impossibility result in Theorem 2.1 of Jacob & Thiery (2015).

# F. Further Details of Experiments in Section 5

## F.1. Heterogeneous Truncated Gaussian

Recall that for fixed $\theta$, our data $\{y_1, y_2, \ldots, y_N\}$ is assumed to be i.i.d. with distribution $y_i \mid \theta \sim \mathbb{N}(\theta, \Sigma)$. With uniform prior on the hypercube $[-K, K]^d$, our target is the following tempered posterior:

$$\pi(\theta \mid y) \propto \exp\left\{-\frac{1}{2}\beta\sum_{i=1}^{N}(\theta - y_i)^\mathsf{T}\Sigma^{-1}(\theta - y_i)\right\}\mathbb{I}(\theta \in [-K, K]^d)$$

Now we look at the function $\tilde{\phi}_i(\theta) = -\frac{1}{2}\beta(\theta - y_i)^\mathsf{T}\Sigma^{-1}(\theta - y_i)$. It is clear that $\tilde{\phi}_i(\theta) \leq 0$. On the other hand:

$$\tilde{\phi}_i(\theta) \geq -\frac{1}{2}\beta\lambda_{\max}(\Sigma^{-1})\|\theta - y_i\|_2^2 \geq -\frac{1}{2}\beta\lambda_{\max}(\Sigma^{-1})\sum_{j=1}^{d}(|y_{ij}| + K)^2,$$

where $\lambda_{\max}(\Sigma^{-1})$ is the largest eigenvalue of $\Sigma^{-1}$. Set

$$M_i := \frac{1}{2}\beta\lambda_{\max}(\Sigma^{-1})\sum_{j=1}^{d}(|y_{ij}| + K)^2,$$

and define $\phi_i(\theta) := \tilde{\phi}_i(\theta) + M_i$, we have $\phi_i(\theta) \in [0, M_i]$ for each $i$. Meanwhile, the posterior is $\pi(\theta \mid y) \propto \exp\{\sum_{i=1}^{N}\phi_i(\theta)\}$, as adding a constant to $\tilde{\phi}_i(\theta)$ does not affect the posterior.

## F.2. Robust Linear Regression

Similarly, we can derive a bound for each

$$\tilde{\phi}_i(\theta) := -\beta \frac{v+1}{2} \log \left( 1 + \frac{(y_i - \theta^\mathsf{T} x_i)^2}{v} \right)$$

in the robust linear regression example. Suppose we constraint $\theta$ in the region $\{\theta, \|\theta\|_2 \leq R\}$, then in order to get the upper bound $M_i$, we first find the following maximum:

$$\max_\theta (y_i - \theta^\mathsf{T} x_i)^2 \quad \text{subject to } \|\theta\|_2 \leq R$$

The above maximization problem can be solved analytically, with the solution:

$$\theta^* = \begin{cases} \frac{-x_i}{\|x_i\|_2} \cdot R & \text{when } y_i > 0 \\[2ex] \frac{x_i}{\|x_i\|_2} \cdot R & \text{when } y_i < 0. \end{cases}$$

The maximum equals $(|y_i| + \|x_i\|_2 \cdot R)^2$ for both cases. Taking

$$M_i = \beta \frac{v+1}{2} \log \left( 1 + \frac{(|y_i| + \|x_i\|_2 \cdot R)^2}{v} \right),$$

and set $\phi_i(\theta) := \tilde{\phi}_i(\theta) + M_i$, we have $\phi_i(\theta) \in [0, M_i]$ for each $i$. Meanwhile, the posterior is $\pi(\theta \mid y) \propto \exp\{\sum_{i=1}^N \phi_i(\theta)\}$.

## F.3. Bayesian Logistic Regression on `MNIST`

For $i = 1, 2, \cdots, N$, let $x_i \in \mathbb{R}^d$ be the features and $y_i \in \{0, 1\}$ be the label. The logistic regression model is $p_\theta(y_i = 1 \mid x) = h(x_i^\top \theta)$, where $h(z) := (1 + \exp(-z))^{-1}$ is the sigmoid function. We assume the data are i.i.d. . With a flat prior, the target posterior distribution is:

$$\pi(\theta \mid y, x) \propto \exp \left\{ -\sum_{i=1}^N -y_i \log h(x_i^\top \theta) - (1 - y_i) \log h(-x_i^\top \theta) \right\}.$$

Let us denote $U_i(\theta) = -y_i \log h(x_i^\top \theta) - (1 - y_i) \log h(-x_i^\top \theta)$. To derive the required bound for $U_i(\theta)$ in Algorithm 6 and 3, first notice that $U_i(\theta)$ is continuous and convex with gradient $\nabla_\theta U_i(\theta) = (h(x_i^\top \theta) - y_i) \cdot x_i$. Then, we have:

$$|U_i(\theta') - U_i(\theta)| \leq \sup_{s \in [0,1]} \|\nabla U_i(\theta + s(\theta' - \theta))\|_2 \|\theta' - \theta\|_2 \leq \|x_i\|_2 \|\theta' - \theta\|_2,$$

where the first inequality follows from mean-value theorem and the second inequality comes from $|(h(x_i^\top \theta) - y_i)| \leq 1$. Thus, we can take $M(\theta, \theta') = \|\theta' - \theta\|_2$ and $c_i = \|x_i\|_2$ for TunaMH and Tuna–SGLD.

# G. Additional Experiments

## G.1. Additional Results on Heterogeneous Truncated Gaussian

## G.2. Additional Experiments on 10-dimensional Robust Linear Regression

Figure 4 compares the MSE over iteration count for all seven algorithms in the robust linear regression example. MALA and Poisson–MALA exhibit the best overall per-iteration performance, followed by HMC, Barker and Poisson–Barker. This indicates that our locally-balanced PoissonMH method can find proposals as effective as full-batch methods while enjoying a very low per-step cost.

Figure 5 compares HMC with 2, 5, and 10 leapfrog steps. Their performances are similar for all acceptance rates, with 10 leapfrog steps being slightly better. Therefore, the HMC used in Section 5.2 uses 10 leapfrog steps.

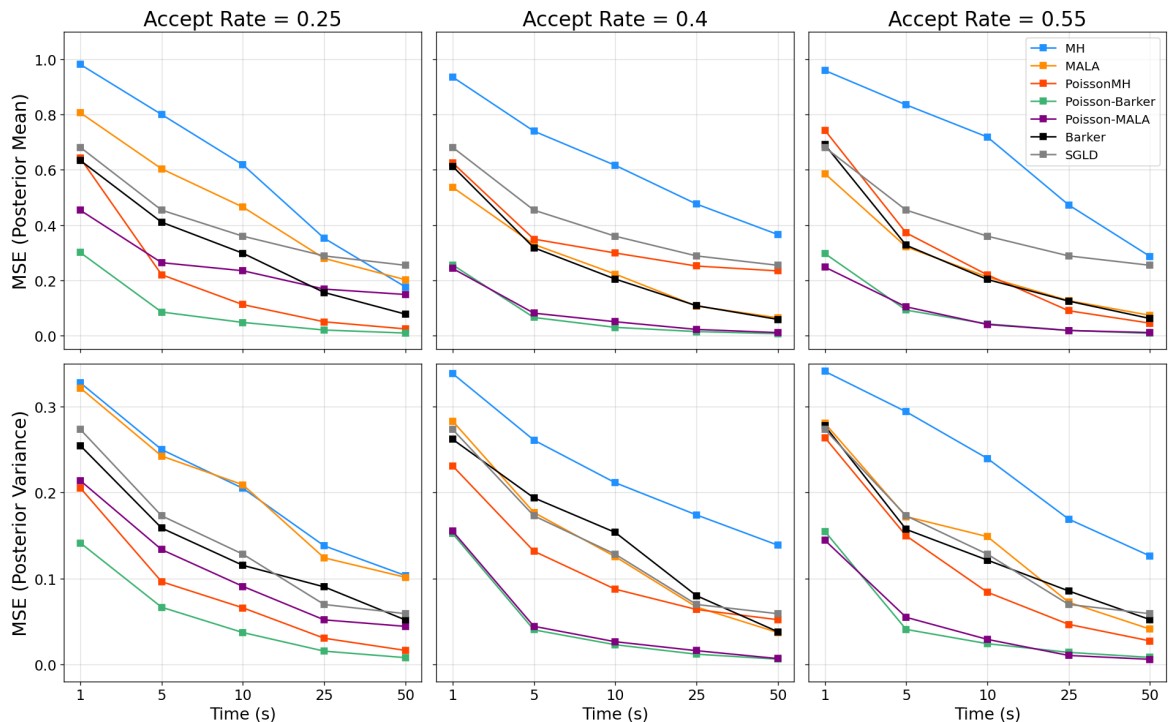

*Figure 3.* Clock-time MSE comparison. First row: MSE of posterior mean as a function of time for different acceptance rates; Second row: MSE of posterior variance as a function of time for different acceptance rates. The results are averaged over 100 runs for all methods.

| | ESS/s: (Min, Median, Max) | | | |
|---|---|---|---|---|
| Method | acceptance rate=0.25 | acceptance rate=0.4 | acceptance rate=0.55 | Best |
| MH | (0.05, 0.08, 0.47) | (0.04, 0.06, 0.44) | (0.03, 0.05, 0.34) | (0.05, 0.08, 0.47) |
| MALA | (0.09, 0.15, 0.81) | (0.10, 0.19, 1.85) | (0.10, 0.19, 2.77) | (0.10, 0.19, 2.77) |
| Barker | (0.11, 0.19, 0.83) | (0.12, 0.22, 1.24) | (0.11, 0.21, 1.53) | (0.12, 0.22, 1.53) |
| PoissonMH | (0.40, 0.66, 4.67) | (0.34, 0.55, 4.51) | (0.27, 0.40, 3.35) | (0.40, 0.66, 4.67) |
| Poisson–Barker | (**0.86**, **1.47**, 6.40) | (**0.91**, **1.65**, 9.75) | (**0.84**, 1.59, 12.16) | (**0.91**, **1.65**, 12.16) |
| Poisson–MALA | (0.77, 1.39, **6.89**) | (0.79, 1.54, **15.14**) | (0.84, **1.65**, **23.84**) | (0.84, **1.65**, **23.84**) |

*Table 4.* ESS/s comparison. Here (Min, Median, Max) refer to the minimum, median and maximum of ESS/s across all dimensions. The column "Best" reports the best ESS/s across all acceptance rates. The results are averaged over 100 runs for all methods.

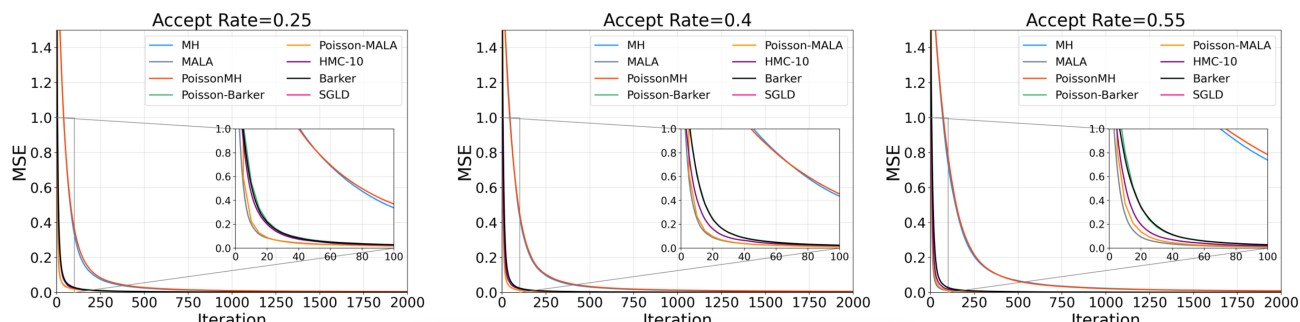

*Figure 4.* Iteration-wise MSE comparison.

We also provide experimental results with a larger data size for the robust regression example. We set $N = 200000$ and the

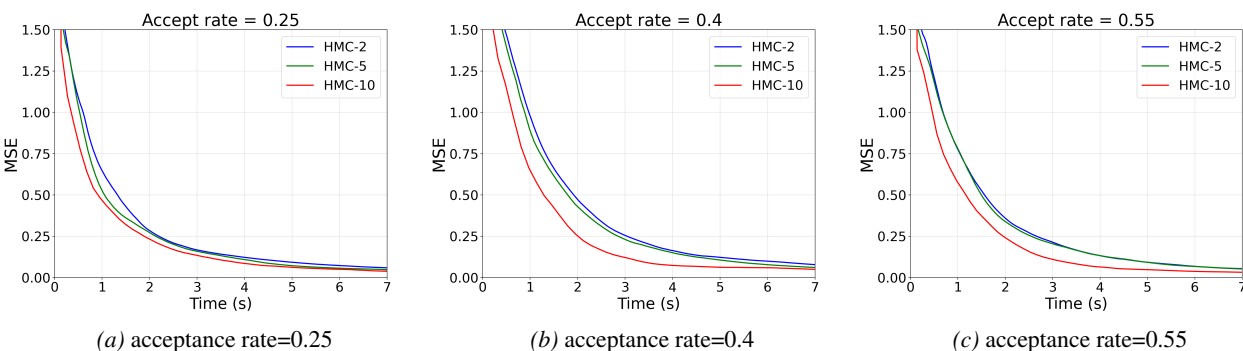

*Figure 5.* Clock-time MSE comparison for HMC with leapfrog steps 2, 5, and 10. All results are averaged over 10 runs.

| | ESS/s: (Min, Median, Max) | | | |
|---|---|---|---|---|
| Method | acceptance rate=0.25 | acceptance rate=0.4 | acceptance rate=0.55 | Best |
| MH | (2.0, 2.1, 2.1) | (1.8, 1.8, 2.0) | (1.2, 1.3, 1.3) | (2.0, 2.1, 2.1) |
| MALA | (2.2, 2.3, 2.4) | (4.3, 4.3, 4.4) | (5.0, 5.1, 5.2) | (5.0, 5.1, 5.2) |
| Barker | (2.0, 2.0, 2.1) | (2.8, 2.8, 2.9) | (2.9, 3.0, 3.0) | (2.9, 3.0, 3.0) |
| HMC-10 | (0.7, 0.8, 0.8) | (0.9, 0.9, 1.0) | (0.8, 0.9, 0.9) | (0.9, 0.9, 1.0) |
| PoissonMH | (99.0, 100.2, 101.1) | (84.9, 87.0, 88.0) | (56.5, 57.9, 59.0) | (99.0, 100.2, 101.1) |
| Poisson–Barker | (182.1, 185.5, 188.2) | (254.8, 258.3, 259.6) | (265.3, 268.3, 270.3) | (265.3, 268.3, 270.3) |
| Poisson–MALA | (**219.8**, **222.0**, **228.5**) | (**395.9**, **399.9**, **403.3**) | (**489.3**, **491.8**, **496.5**) | (**489.3**, **491.8**, **496.5**) |

*Table 5.* ESS/s comparison. Here (Min, Median, Max) refer to the minimum, median and maximum of ESS/s across all dimensions. The column "Best" reports the best ESS/s across all acceptance rates. The results are averaged over 100 runs for all methods.

other settings remain the same. The MSE vs time/iteration comparison is summarized in Figure 6, and the ESS/s comparison are summarized in Table 6.

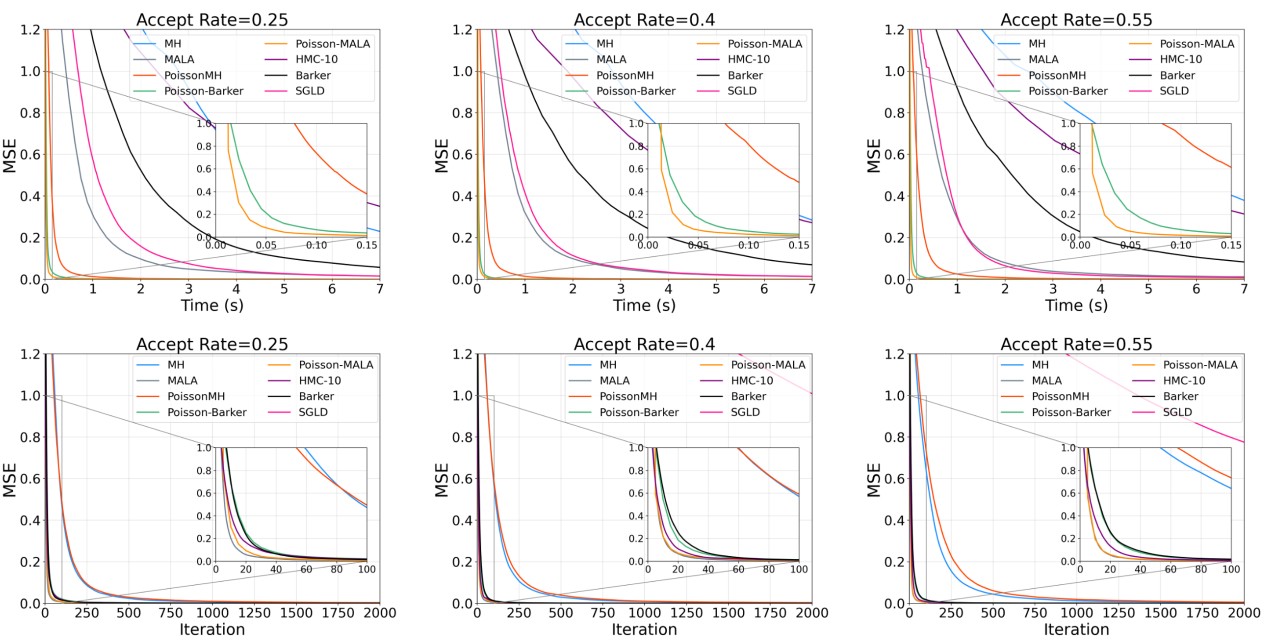

*Figure 6.* Clock-time and iteration-wise MSE comparison.

*Table 6.* ESS/s comparison across target acceptance rates.

| Method | ESS/s: (Min, Median, Max) | | | |
| --- | --- | --- | --- | --- |
| | target rate=0.25 | target rate=0.4 | target rate=0.55 | Best |
| MH | (0.61, 0.64, 0.70) | (0.55, 0.61, 0.68) | (0.41, 0.46, 0.51) | (0.61, 0.64, 0.70) |
| MALA | (0.83, 0.87, 0.93) | (1.26, 1.35, 1.42) | (1.48, 1.56, 1.65) | (1.48, 1.56, 1.65) |
| HMC5 | (0.21, 0.22, 0.24) | (0.22, 0.24, 0.26) | (0.21, 0.23, 0.26) | (0.22, 0.24, 0.26) |
| Barker | (0.53, 0.59, 0.61) | (0.68, 0.73, 0.80) | (0.75, 0.79, 0.81) | (0.75, 0.79, 0.81) |
| PoissonMH | (24.67, 25.74, 26.86) | (21.38, 22.16, 22.97) | (14.17, 15.31, 16.22) | (24.67, 25.74, 26.86) |
| Poisson–Barker | (39.32, 40.57, 41.88) | (56.82, 58.72, 60.88) | (58.40, 60.33, 61.33) | (58.40, 60.33, 61.33) |
| Poisson–MALA | (52.68, 54.31, 57.42) | (89.36, 92.04, 94.31) | (109.03, 111.01, 114.40) | (109.03, 111.01, 114.40) |

### G.3. 50-dimensional Robust Linear Regression

We compare Poisson–Barker, Poisson–MALA, PoissonMH, random-walk Metropolis, MALA, Barker and HMC in a 50-dimensional robust linear regression example. All setups are the same as in Section 5.2, except for $d = 50$, and the number of leapfrog steps in HMC is 5.

Figure 7 shows the MSE of estimating the true coefficients over time. Figure 8 shows the MSE comparison over iteration count. Table 7 compares the ESS/s. Figure 9 compares HMC with 2, 5, and 10 leapfrog steps. The higher dimensionality slows the mixing of all algorithms compared to Section 5.2, but all our conclusions remain valid. Our Poisson–MALA and Poisson–Barker outperform all other algorithms.

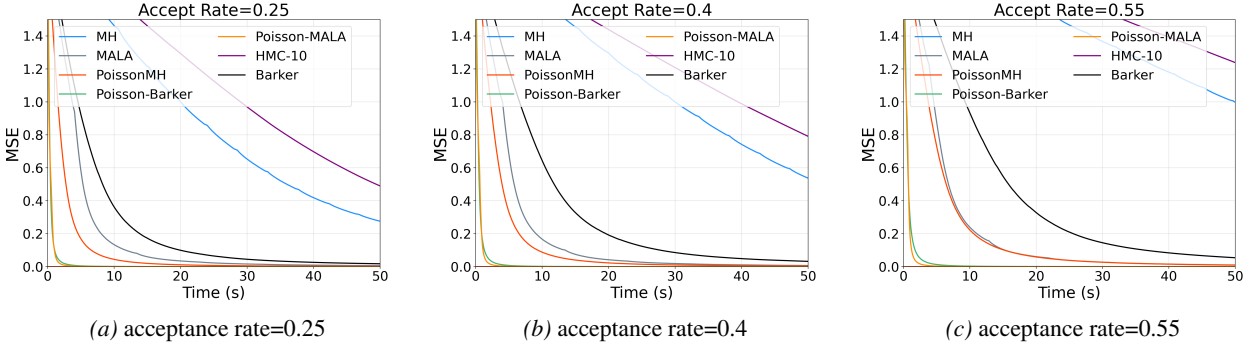

*(a)* acceptance rate=0.25     *(b)* acceptance rate=0.4     *(c)* acceptance rate=0.55

*Figure 7.* Clock-time MSE comparison. MSE of $\theta^\star$ as a function of time across different acceptance rates. The three large plots show the performance of all methods in the first 50 seconds.

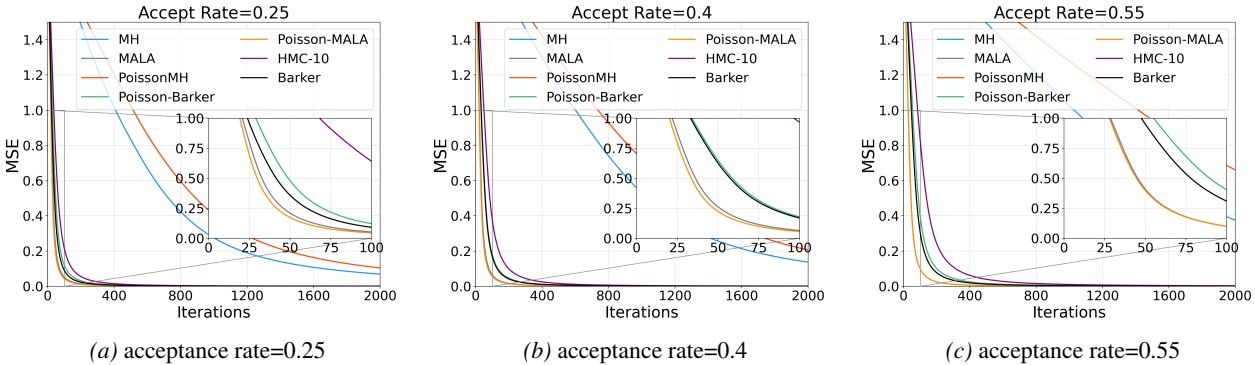

*(a)* acceptance rate=0.25     *(b)* acceptance rate=0.4     *(c)* acceptance rate=0.55

*Figure 8.* Iteration-wise MSE comparison. MSE of $\theta^\star$ as a function of iteration count different acceptance rates. The three large plots show the performance of all methods in the first 2000 steps. The three inside plots zoom in on Poisson–Barker, and Poisson–MALA, MALA, and HMC in the first 100 steps. All results are averaged over 10 runs.

| Method | ESS/s: (Min, Median, Max) | | | |
| | acceptance rate=0.25 | acceptance rate=0.4 | acceptance rate=0.55 | Best |
|---|---|---|---|---|
| MH | (0.08, 0.10, 0.13) | (0.07, 0.08, 0.12) | (0.06, 0.07, 0.08) | (0.08, 0.10, 0.13) |
| MALA | (0.39, 0.48, 0.56) | (0.63, 0.72, 0.81) | (0.73, 0.83, 0.91) | (0.73, 0.83, 0.91) |
| Barker | (0.17, 0.21, 0.25) | (0.23, 0.26, 0.29) | (0.23, 0.28, 0.31) | (0.23, 0.28, 0.31) |
| HMC-5 | (0.03, 0.04, 0.05) | (0.03, 0.03, 0.04) | (0.02, 0.03, 0.04) | (0.03, 0.04, 0.05) |
| PoissonMH | (1.91, 2.04, 2.21) | (1.57, 1.66, 1.83) | (1.24, 1.34, 1.44) | (1.91, 2.04, 2.21) |
| Poisson–Barker | (4.18, 4.54, 4.78) | (5.47, 5.85, 6.29) | (6.07, 6.51, 6.84) | (6.07, 6.51, 6.84) |
| Poisson–MALA | (**6.76**, **7.45**, **8.11**) | (**11.00**, **11.81**, **12.40**) | (**13.92**, **14.91**, **15.58**) | (**13.92**, **14.91**, **15.58**) |

*Table 7.* ESS/s comparison. Here (Min, Median, Max) refer to the minimum, median and maximum of ESS/s across all dimensions. The column "Best" reports the best ESS/s across all acceptance rates. The results are averaged over 10 runs for all methods.

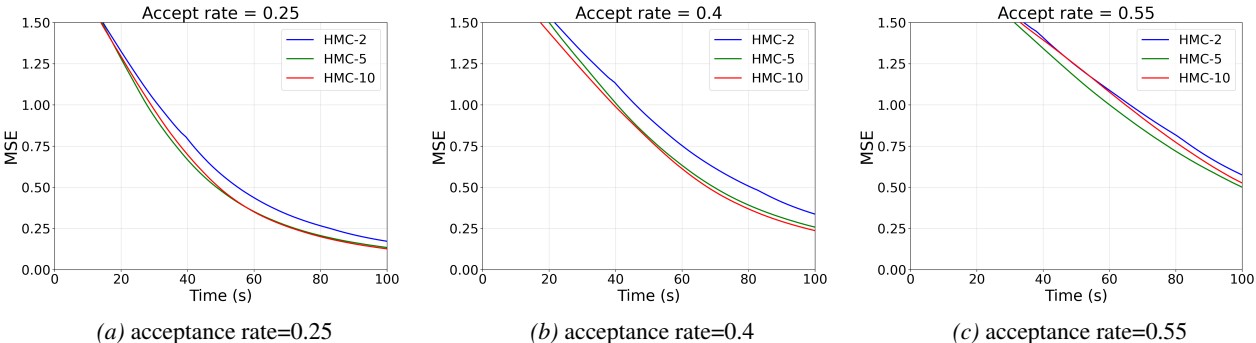

*(a) acceptance rate=0.25*     *(b) acceptance rate=0.4*     *(c) acceptance rate=0.55*

*Figure 9.* Clock-time MSE comparison for HMC with leapfrog steps 2, 5, and 10. All results are averaged over 10 runs.

### G.4. Additional Experiments on Bayesian Logistic Regression: 3s and 5s

We compare all algorithms, along with an additional HMC with 2 leapfrog steps, over a broader range of step sizes, some of which are much larger than those used in (Zhang et al., 2020). The test accuracy for each algorithm with varying step sizes is summarised in Figure 10. Increasing the step size overall improves the performance of every full-batch algorithm, especially for HMC. However, the plot clearly shows that the HMC algorithm still struggles with mixing, even with very large step sizes, typically taking more than 30 seconds to converge.

Figure 11 shows a comparison of all algorithms when each is optimally tuned. From the left panel, it is clear that Tuna–SGLD converges fastest among all algorithms, followed by TunaMH and random-walk Metropolis. The right panel indicates that MALA exhibits superior iteration-wise convergence compared to Tuna–SGLD, while random-walk Metropolis demonstrates better iteration-wise convergence than TunaMH. This observation supports our theoretical findings in Section 4.1.

To provide further guidance on the hyperparameter tuning for our new algorithm Tuna–SGLD, we vary the minibatch size for stochastic gradient in 20, 64, 128, and try different step sizes. The results are provided in Figure 12. The performance of Tuna–SGLD is not sensitive to step sizes, unless step size is large. This is because the minibatch size of Tuna–SGLD is larger when the step size is larger. We recommend using $10^{-5}$ as suggested in (Zhang et al., 2020). For the minibatch size of estimating the gradient, we use a relatively small choice (size=20), since it is sufficient for Tuna–SGLD to converge fast and using size=64/128 will bring additional computational cost.

### G.5. Bayesian Logistic Regression: 7s and 9s

Similar to Section 5.3, we test TunaMH, Tuna–SGLD, random–walk Metropolis, MALA, Barker and HMC on Bayesian logistic regression task to classify 7s and 9s in the MNIST data set. The training set contains 12,214 samples, and the test set contains 2,037 samples. We adopt all implementation details from Section 5.3 and report the results in Figure 13. We can draw similar conclusions to Section 5.3 that our proposed minibatch gradient-based algorithm performs better than its

original minibatch non-gradient version in all step size configurations. All minibatch methods still significantly improve the performances of full-batch methods.

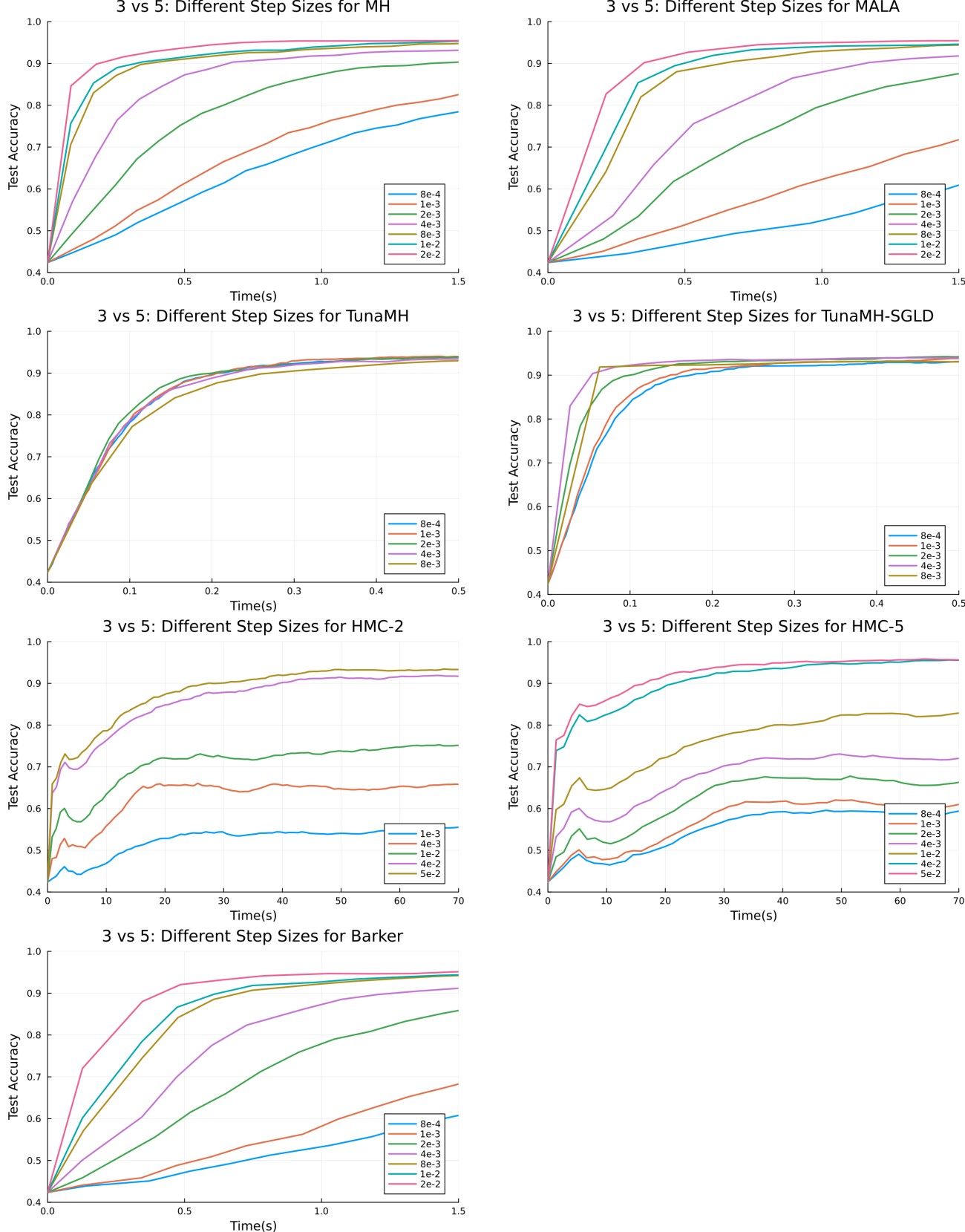

*Figure 10.* Test accuracy as a function of time for each algorithm with different step sizes.

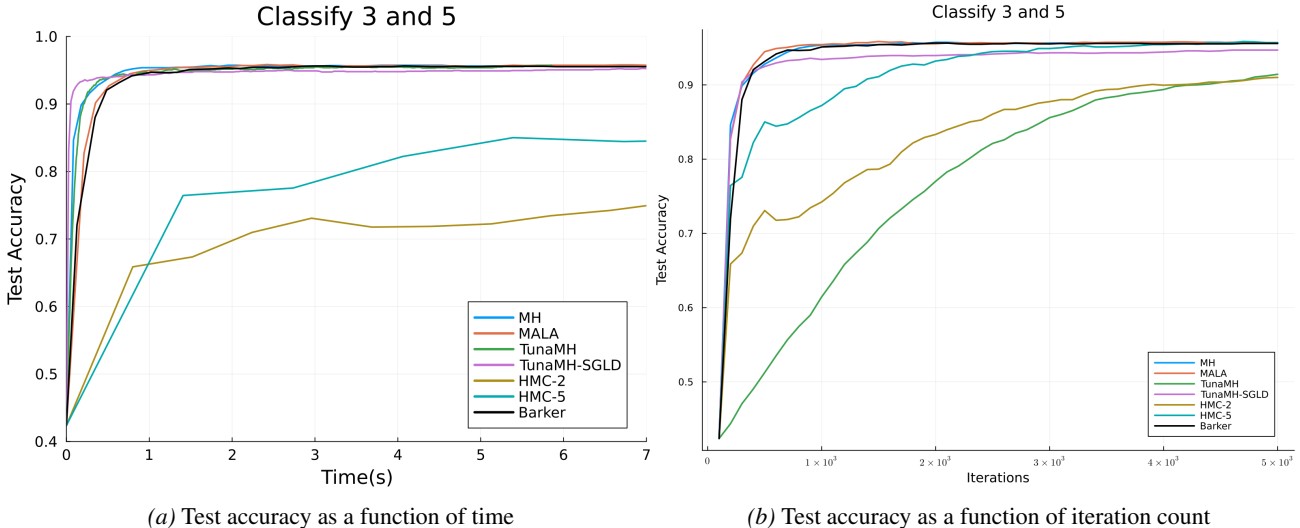

*(a)* Test accuracy as a function of time                    *(b)* Test accuracy as a function of iteration count

*Figure 11.* Test accuracy comparison when each algorithm is optimally tuned.

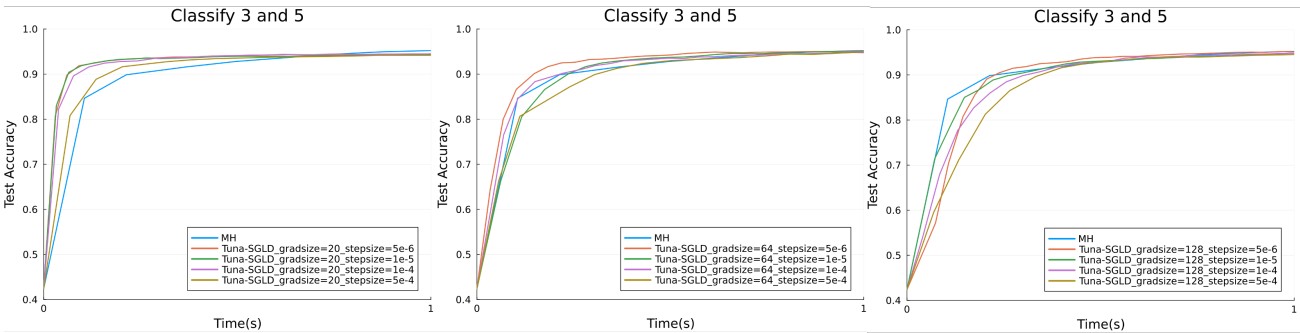

*Figure 12.* Test accuracy of Tuna–SGLD as a function of time for classifying 3s and 5s in `MNIST`, across different gradient minibatch sizes and step sizes.

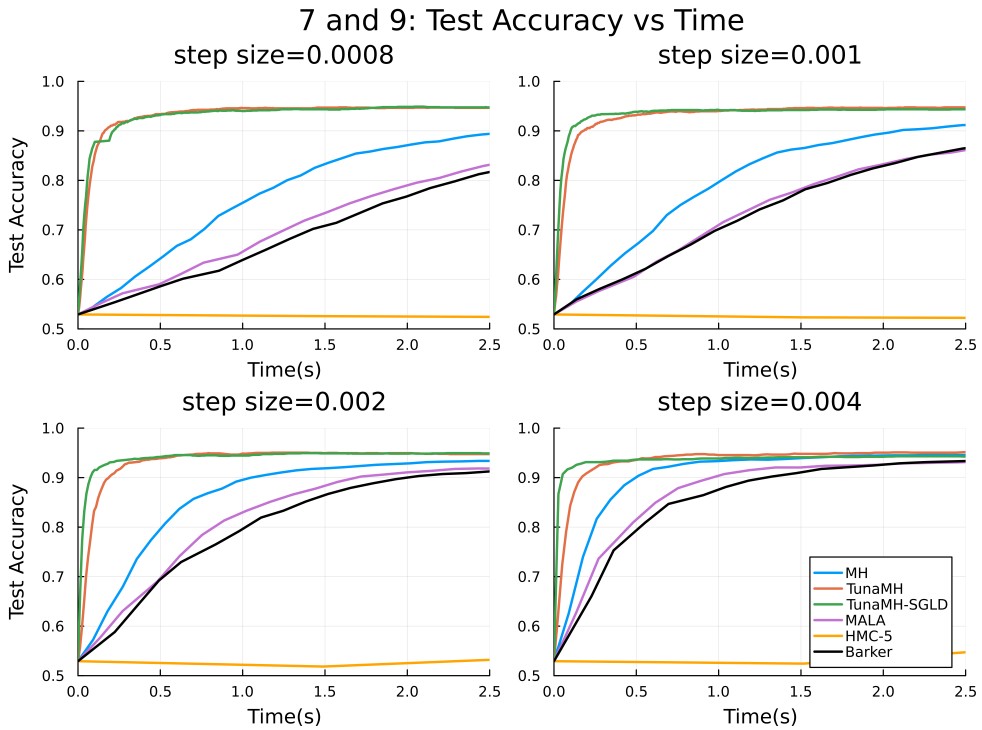

*Figure 13.* Test accuracy as a function of time for classifying 7s and 9s in `MNIST`. Step sizes for all methods are varied across $\{0.0008, 0.001, 0.002, 0.004\}$. Each subplot reports the curve of test accuracy in the first 2.5 seconds.

