# OpenReview forum: "Markov Chain Monte Carlo without Evaluating the Target: an Auxiliary Variable Approach"
_ICML.cc/2026/Conference — ICML 2026 spotlight_

### Official Review · Reviewer_fGYJ · 2026-03-10

**Soundness:** 3
**Presentation:** 2
**Significance:** 3
**Originality:** 3
**Overall Recommendation:** 5
**Confidence:** 2

**Summary:**

This paper shows that several existing MCMC methods can be unified under a broader meta-MCMC framework. Given this insight, they propose a gradient-based MCMC that does not evaluate the full posterior. As they mention, their algorithm does not suffer from a bias that affects some existing minibatch gradient-based MCMC approaches.  They also provide a theoretical analysis of their proposed method.

**Compliance With Llm Reviewing Policy:**

Affirmed.

**Key Questions For Authors:**

1. In Algorithm 1, line 4, why is the proposal joint, i.e. $q(\theta_t, \cdot)$ rather than conditional, i.e., $q(\cdot | \theta_t)$?

2. In page1, I suggest, instead of the term "derivative of the unnormalized distribution", use the term "gradient of ..." to highlight the multivariable nature of the problem.

**Limitations:**

This work is theoretical with no direct negative societal impact.

**Strengths And Weaknesses:**

Strengths:
As far as I followed the proofs, their theoretical findings seem sound; the main problem that they tackle, namely, proposing a gradient-informed unbiased MCMC sampler for "tall dataset" settings, is innovative and significant. Also, unifying several approaches under a larger umbrella leads to a valuable insight and a better understanding of the links between these approaches and has the potential to pave the way to future samplers, other than the samplers proposed in this work; therefore, it is a valuable contribution.

Weakness:
1. The presentation can be improved. For instance, in the beginning of Section 2, they suddenly refer to Algorithms 4-6 in the appendix without introducing and motivating them first.
2. Their experimental results are averaged over 10 runs only, making it sensitive to the initial state. Given that each experiment takes only 7 seconds, they could easily average over hundreds of runs to reduce this initial dependency and provide a fairer comparison.
3. The importance of their theoretical findings could be further discussed.

---

> ### Author Rebuttal · Authors · 2026-03-30
>
> Thank you very much for your postive review. We are glad to know that you find the contribution of our work valuable. We address your comments below.
>
> **(1) Experiment Results Averaged over More Runs**
>
> Great point. We provide experiment results averaged over 100 independent runs for the truncated Gaussian example and the 10-dimensional robust regression example. We provide the results in the [anomynous github repo](https://anonymous.4open.science/r/Auxiliary-MCMC-ICML2026-5ECD). Please refer to Figure 1 & 2 and Table 1 & 2. Briefly, the conclusions are consistent with what we have stated in the paper. Our methods converge much faster than PoissonMH and other full-batch methods. For truncated Gaussian example, Poisson-MALA and Poisson-Barker improve PossionMH by 1.4 - 7.1 times in ESS/s, improve full-batch methods by 6.1 - 77.1 times in ESS/s. For robust regression example, Poisson-MALA and Poisson-Barker improve PossionMH by 1.8 - 8.7 times in ESS/s, improve full-batch methods for nearly over 100 times in ESS/s.
>
> **(2) Discussion on the Importance of Our Theoretical Finding**
>
> There are two valuable aspects of our theoretical result. First, it improves on existing results, while relying on arguments that are arguably simpler and more transparent. This makes the guarantees easier to check and may also make the proof ideas more useful in related problems. Second, our theoretical framework gives a general way to use existing comparison results for MCMC algorithms to obtain explicit guarantees in these settings. Since MCMC comparison theory has developed rapidly in recent years, our result serves both as a proof of concept for applying this line of theory in a concrete problem and as a natural motivating example for further work in that area.
>
> We will add a more detailed discussion on our theoretical aspects of our paper in the revised version.
>
> **(3) Presentaion of Section 2 and Terminology**
>
> Thanks for the comment! We agree that the transition at the beginning of Section 2 could be made clearer. We will revise this part to better introduce and motivate these algorithms before referring to them, and ensure a smoother presentation in the final revision.
>
> Thank you for pointing out that we should use “gradient” instead of “derivative” for multivariate problems. We have revised this sentence.
>
> **(4)  Clarification of Proposal in Algorithm 1**
>
> Thank you for the clarification. We have added a sentence to explicitly state that $q(\theta_t, \cdot)$ denotes a transition kernel (i.e., a conditional distribution given the current state $\theta_t$), rather than a joint proposal.

---

> > ### Author Rebuttal · Reviewer_fGYJ · 2026-04-03
> >
> > I thank the authors for carrying out new experiments which surely adds to the value of the paper. I keep my original score which is "accept".

---

### Official Review · Reviewer_XnjG · 2026-03-11

**Soundness:** 4
**Presentation:** 4
**Significance:** 3
**Originality:** 3
**Overall Recommendation:** 5
**Confidence:** 4

**Summary:**

This work presents a rigorous unifying framework for exact subsampling and doubly-intractable likelihood in MCMC, elegantly synthesizing disparate algorithms like PoissonMH, TunaMH, and the Exchange algorithm under a novel dual-auxiliary variable meta-algorithm. The work’s most significant contribution is the derivation of new gradient-based minibatch algorithms, locally balanced poissonMH and Tuna-SGLD, providing a principled, exact solution for debiasing MCMC based on minibatch gradient information.

**Compliance With Llm Reviewing Policy:**

Affirmed.

**Key Questions For Authors:**

I'm curious, is it possible to cast the dual-auxiliary framework into the broad involutive MCMC framework, e.g., composition of two commuting involutions? Concretely, I'm thinking involution A as the proposal swap $$F_A\Big( \theta, \theta', \omega_1, \omega_1', \omega_2, \omega_2' \Big) = \Big( \theta', \theta, \omega_1', \omega_1, \omega_2, \omega_2' \Big)$$, and involution B as the pseudo-marginal swap: $$F_B\Big( \theta, \theta', \omega_1, \omega_1', \omega_2, \omega_2' \Big) = \Big( \theta, \theta', \omega_1, \omega_1', \omega_2', \omega_2 \Big)$$


This would drastically simplify the justification of the detail balance condition.

**Limitations:**

I think this work would be strengthened by adding a dedicated limitations paragraph acknowledging:
- The pre-computation and memory overhead required to construct an alias table for the entire dataset.
- The proposed algorithms require substantial hyperparameter tuning

**Strengths And Weaknesses:**

Strength:

- I appreciate the authors clearly writing and vcomprehensive literature review and a clearly articulated statement of contributions.

- In my opinion, I enjoy the presentation of the dual-auxiliary variable framework, which provides an elegant, unifying interface for several seemingly disparate algorithms. The discussion of the three specific cases in Section 3.1 serves as a particularly clear illustration of this unification.

- The work provides rigorous theory establishing the reversibility of the dual-auxiliary variable framework. The authors cleverly utilize Peskun’s ordering to compare the transition density and spectral gap of the proposed auxiliary chain against idealized and Metropolis-within-Gibbs chains. This unifying framework offers a convenient platform for comparing and evaluating different MCMC algorithms.

- Arguably, the most significant contribution of this work lies in the derivation of two novel algorithms from the framework: Locally Balanced PoissonMH (incorporating Barker and MALA proposals) and Tuna-SGLD (a Metropolized version of Stochastic Gradient Langevin Dynamics). To my knowledge, this provides a principled and exact solution to the long-standing open problem of correcting SGLD's systematic bias using a Metropolis-Hastings step based strictly on minibatches. This is a substantial step forward compared to previous methods---such as step-size annealing or control variates---which mitigate but do not fully eliminate bias.

- The new gradient-based algorithms demosntrate quite significant performance gain than existing methods.

Weakness:

- The proposed algorithms require substantial tuning: step size, minibatch size, balancing functions, bounding constants, abd the parameter $\chi$. The paper does not provide robust automated tuning strategies, which could be burden for practioners. However, I wish to emphasize that this practical limitation does not outweigh the significant theoretical and algorithmic contributions of the work.

-  The empirical evaluations are largely constrained to relatively simple target distributions. Furthermore, the dataset sizes for the robust linear regression ($10^5$) and Bayesian logistic regression (~11K) tasks may not fully represent true "tall data" scenarios. I strongly recommend evaluating these algorithms on larger-scale problems with significantly taller datasets (e.g., those utilized in Coreset Markov Chain Monte Carlo, Chen & Campbell, 2024) to better demonstrate their scalability.

- Unless I missed it, the paper does not explicitly specify the test functions used for calculating the Effective Sample Size (ESS) in the experiments.

Minor issue:

- Missing classical literature on pseudo-marginal MCMC: Andrieu & Roberts (2009), The pseudo-marginal approach for efficient Monte Carlo computations, AOS.

- It's worth including a discussion on those PDMP-based (piecewise deterministic Markov process) exact subsampling methods.Specifically, the Bouncy Particle Sampler (Bouchard-Côté et al., 2018) and the Zig-Zag process (Bierkens et al., 2019) are major competing methods for exact inference on tall data.

---

> ### Author Rebuttal · Authors · 2026-03-31
>
> Thank you for your positive reviews! We are encouraged that you find our contribution in the framework and algorithms significant, our presentation enjoyable, and writing clear. We address your comments below.
>
> **(1) Hyper-paremeter Tuning**
>
> Thank you for your comment. We agree that our methods require some tuning, and we will add a paragraph discussing this limitation in our final revison. We hereby provide some general heuristics for choosing the hyperparameters, which we find to work well across our experiments.
>
> For step size tuning of Possion-Barker and Poisson-MALA, we adopt the strategy to run some pre-experiments to reach a given target accept rate. In practice, users can also adaptively adjust the step size to reach a target accept rate. For the choice of minibatch size for Poisson-MALA and Poisson-Barker, we take the hyper-parameter $\lambda=c\cdot L^2$ as suggeted in [1]. The balancing functions are chosen as $g(t) = t/(1+t)$ and $g(t) = \sqrt{t}$, which corresponds to Barker’s algorithm and MALA. The bounding constants depend on the model and the dataset, with detalied derivation in Section F in the appendix.
>
> For the effect of hyper-parameter of Tuna-SGLD, we conduct additional experiments on the MNIST dataset (see Figure 4 in the [anomynous github repo](https://anonymous.4open.science/r/Auxiliary-MCMC-ICML2026-5ECD)). We vary the minibatch size for stochastic gradient in {20, 64, 128}, and try different step sizes. The performance of Tuna-SGLD is not sensitive to step sizes, unless step size is large. This is because the minibatch size of Tuna-SGLD is larger when the step size is larger. We recommend using 1e-5 as suggested in [2]. For the minibatch size of estimating the gradient, we use a relatively small choice (size=20), since it is sufficient for Tuna-SGLD to converge fast and using size=64/128 will bring additonal computational cost.
>
> **(2) Evaluation on Larger-scale Dataset**
>
> Thanks a lot for pointing this out. We further test our methods on the robust regression example using 200000 data points (the dateset size used in the paper is 100000), see Table 3 and Figure 3 in the [anomynous github repo](https://anonymous.4open.science/r/Auxiliary-MCMC-ICML2026-5ECD). The results remain consistent with our previous experiments. Our methods significantly outperforms full-batch methods, and outperform mini-batch methods without gradient. It would also be interesting to apply our methods to the datasets used in [3], which we leave for future work.
>
> **(3) Involutive MCMC perspective**: After thinking about it carefully, we think Algorithm 1 has a clean involutive-MCMC interpretation. Our view is to treat all forward randomness generated in Steps 3–5 as a single auxiliary variable $v$, with conditional density $m_\theta(v) = P_\theta(\omega_1)q_{\omega_1}(\theta,\theta') P_{\theta,\theta'}(\omega_2\mid \omega_1)$, and the involution $f(\theta, \omega_1, \theta',\omega_2) = (\theta', \omega_1, \theta,\omega_2)$. Then the resulting involutive-MCMC acceptance ratio is precisely Algorithm 6 of our algorithm. Hence the validity proof can be viewed as an application of involutive MCMC validity.
>
> We appreciate the insightful comment, and will add a detailed discussion in our revised version.
>
> **(4) Discussion on Related Work**
>
> Thanks for mentioning these, we will discuss these in the revised version.
>
> **(5) Computation and memory-overhead of Alias Table and Test Functions for ESS**
>
> We agree that there is a pre-computation cost and memory overhead associated with the alias table. However, this cost is incurred only once and can be amortized over the entire sampling procedure. The memory overhead of the alias table is O(N), consisting of approximately two arrays of dataset size, which is modest in practice. We will clarify this trade-off in our final revision.
>
> We apply a standard autocorrelation-based ESS calculation to each dimension of the parameter, which corresponds to using the identity function as the test function. We evaluate all methods using the minimum, median, and maximum ESS/s across all dimensions.
>
> [1] Zhang, R. and De Sa, C. M. Poisson-minibatching for gibbs sampling with convergence rate guarantees. Advances in Neural Information Processing Systems, 2019.
>
> [2] Zhang, R., Cooper, A. F., and De Sa, C. M. Asymptotically optimal exact minibatch Metropolis–Hastings. Advances in Neural Information Processing Systems, 2020
>
> [3] Chen N, Campbell T. Coreset Markov chain Monte Carlo. InInternational Conference on Artificial Intelligence and Statistics 2024

---

> > ### Author Rebuttal · Reviewer_XnjG · 2026-04-02
> >
> > The authors have fully addressed my concern, and I am maintaining my positive evaluation to this work.

---

### Official Review · Reviewer_X2nJ · 2026-03-11

**Soundness:** 3
**Presentation:** 3
**Significance:** 3
**Originality:** 2
**Overall Recommendation:** 5
**Confidence:** 4

**Summary:**

This paper first observed a unified procedure among a variety of existing MCMC algorithms, including the exchange algorithm, PoissonMH, and TunaMH, that has been designed for contexts where evaluating the target distribution is costly or not feasible. In their general auxiliary-variable MCMC framework, that combines two auxiliary variables—one for designing the proposal distribution and the other for estimating the target density ratio in the acceptance step—is based on this observation. A theoretical analysis of the resulting Markov chains is provided. Within this framework, the paper further develops new gradient-based minibatch MCMC algorithms, in which more efficient proposal distributions are generated using minibatch gradient estimates. Numerical experiments demonstrate improved sampling efficiency of the proposed algorithms compared with a variety of existing algorithms.

**Compliance With Llm Reviewing Policy:**

Affirmed.

**Final Justification:**

The rebuttal addressed all of my main concerns, and I therefore raise my score to accept.

**Key Questions For Authors:**

(1) Could the authors explain on how the experiments' baseline methods—particularly HMC—were tuned?

(2) Are there any practical suggestions/guidelines for selecting the minibatch parameters or auxiliary distributions before running it in practical applications?

(3) Could you clarify on how your method differs from current pseudo-marginal frameworks?

**Limitations:**

The proposed framework relies on stochastic estimators constructed using minibatches and auxiliary variables. The efficiency of such algorithms may depend on the variance of these estimators, but the paper does not provide much analysis of how sensitive the methods are to estimator noise.

**Strengths And Weaknesses:**

Strengths:

(1) This paper presents a general auxiliary-variable framework that could be used with various existing MCMC algorithms, including TunaMH, PoissonMH, and the exchange algorithm. This unification offers a clearer pattern of how these approaches are related and is conceptually elegant.

(2) The paper establishes connections with current algorithms and presents theoretical results analysing the transition kernels of the proposed framework.

(3) The paper compares several algorithms and reports various performance metrics using three examples on both synthetic and real datasets.

Weaknesses:

(1) Baseline comparisons may not be fully convincing. It is concerning that the ESS values reported for HMC in experiments are very low. So, I am wondering about whether the algorithm was adequately tuned or given enough time to converge in the tested examples, since HMC is usually a strong baseline for Bayesian inference problems. For instance, the MSE curves for HMC in Figures 5 and 6 (robust linear regression with 50 dimensions) indicate that the chain might not have completely converged within the stated runtime. The test accuracy of HMC in the Bayesian logistic regression experiment is likewise extremely low and essentially stays constant over time, which may suggest that the method did not converge during the experiment. In addition, the paper does not provide convergence diagnostics or tuning details for HMC or other baseline methods. If the baselines are not well-tuned or given sufficient runtime, the improvements of the proposed methods may be less convincing.

(2) Missing baseline comparisons with stochastic gradient MCMC. It would natural to compare the suggested method with some stochastic gradient MCMC algorithms (such as SGLD). A better understanding of how the proposed methods compare to its closely related baseline algorithm could support the paper better.

(3) Relationship to pseudo-marginal MCMC could be clarified. The proposed framework appears closely related to pseudo-marginal MCMC methods[1], which also rely on stochastic estimators within the Metropolis–Hastings framework. The novelty here seems to diminish by this. A clearer discussion of how the proposed framework differs conceptually from existing pseudo-marginal approaches would help better define the novelty and contribution for this work.

[1] Andrieu C, Roberts GO. The pseudo-marginal approach for efficient Monte Carlo computations.

---

> ### Author Rebuttal · Authors · 2026-03-30
>
> Thank you very much for your constructive feedbacks! We address your questions and comments below.
>
> **(1) Hyper-parameter Tuning of Baseline Methods**
>
> Good point. We provide the tuning details of baseline methods, particularly HMC.
>
> For robust regression example, we tune the step size of each algorithm to a target accept rate {0.25, 0.4, 0.55} by a few pre-experiments. We compare the performance of different algorithms across different accept rates. For HMC, we additionally try to tune the leapfrog steps across {2, 5, 10} and take the leapfrog step with best performance. For example, in the 50-dimensional robust regression experiment, we take 5 leapfrog steps since it overall converges faster than 2 or 10 leapfrog steps, see Figure 7. All algorithms are given enough steps to converge. Figure 6 shows MSE vs iterations for all algorithms, and they converge in 2000 iterations except for MH and PoissonMH. Since we want to compare the clock-wise performance, and for a better presentation of the results, we take the first 50 seconds in Figure 5. Full-batch methods requires significantly more time than mini-batch methods per step, and HMC required additional leapfrog steps, so they seem not convergent in Figure 5.
>
> For Bayesian logistic regression, we also tune HMC’s number of leapfrog step and step size. We try leapfrog steps across {2, 5} and several different choices of step sizes. As shown in Figure 8, HMC does converge to test accuracy at 0.95, but requires about 70 seconds due to full-batch date evaluation and additonal leapfrog steps. For clear presentation of the results, we only take the first 1.5 seconds in Figure 2.
>
> **(2) Comparison with SGLD**
>
> Great point. We provide additional comparison with SGLD in robust regression and truncated Gaussian examples. The results are provided in the [anomynous github repo](https://anonymous.4open.science/r/Auxiliary-MCMC-ICML2026-5ECD), see Figure 1 & 2. Briefly, SGLD has a much lower per-iteration cost than the other methods because it avoids accept-rejection. However, this also introduces systematic bias in sampling, which hurts estimation since the bias does not vanish over time.
>
> In robust regression, our minibatch gradient-based methods, Poisson-MALA and Poisson-Barker, outperforms SGLD significantly, see Figure 1. For truncated Gaussian example, similar results are observed in Figure 2. In both cases, although SGLD adapts to the posterior fast in the early stage, its performance flattens between 25 and 50 seconds. This comes from its systematic bias. We further support this claim by Kolmogorov-Smirnov statistics in Table 4 of our repo. The KS statistic of SGLD is between 0.06 and 0.40, which means it adapts to some dimensions very well and mixes very poorly on other dimensions.
>
> **(3) Comparison with pseudo-marginal approach**
>
> Thank you for this question! There are connections with pseudo-marginal MCMC, but our method is not just a rephrasing of the current pseudo-marginal framework. Indeed, one of the earliest paper of minibatch MCMC (FireFly MC ) [1] belongs to pseudo-marginal MC. In the minibatch MC area, pseudo-marginal MC algorithms are often called 'stateful' methods, where our method as well as PoissonMH and TunaMH are called 'stateless' method.
>
> In the standard pseudo-marginal framework, one typically constructs an exact MH algorithm on an **augmented space** $(\theta, U)$ using a nonnegative unbiased estimator of the target or likelihood. By contrast, our framework defines MCMC on $\theta$, but uses a pair of auxiliary variables to guide the move and estimate the acceptance ratio. We do not currently see a direct way to place our algorithm on such an augmented space. A straightforward augmentation by $(\theta,\omega_1,\omega_2)$ would make the acceptance-rejection step depend on the auxiliary variable from the previous iteration, which is not how our algorithm operates.
>
>
>
> We will add more detailed discussion on pesudo-marginal MCMC in our revised paper.
>
> **(4) Choice of Minibatch Parameters**:
>
> Thanks for the question. Due to space constraints, we only give a brief explanation here; more details and additional experiments are provided in our response to Reviewer XnjG. At a high level, tuning amounts to finding a balance between estimation accuracy and per-iteration cost. Using a smaller minibatch makes each iteration faster, but also increases the variance of the estimator.
>
> For Poisson-Barker and Poisson-MALA, we tune the step size through pilot runs to target a reasonable acceptance rate; in practice, adaptive tuning can also be used. We set the remaining hyper-parameters following the appendix of PoissonMH.
>
> For Tuna-SGLD, additional MNIST experiments in our repo show that the method is fairly robust to hyper-parameter choices, except when the step size is too large. In practice, we use the setting recommended in TunaMH.
>
> [1] Firefly Monte Carlo: Exact MCMC with subsets of data. Mclaurin and Adams, UAI.

---

> > ### Author Rebuttal · Reviewer_X2nJ · 2026-04-03
> >
> > The rebuttal addressed all my main concerns. I really appreciate the additional experimental comparisons provided in such a short time, as they clearly required substantial effort, and the new comparison with SGLD strengthens the empirical evaluation. I therefore raise my score to accept.

---

### Official Review · Reviewer_4qKg · 2026-03-12

**Soundness:** 4
**Presentation:** 3
**Significance:** 3
**Originality:** 3
**Overall Recommendation:** 5
**Confidence:** 4

**Summary:**

This paper presents a unifying framework for sampling from target distributions using MCMC in cases where the target/gradient evaluation can be particularly computationally expensive or intractable. In particular, the authors propose a framework that unifies PoissonMH, TunaMH, and the exchange algorithm, among other algorithms. From this, new methods are proposed and studied in the experiments, with desirable performance properties.

**Compliance With Llm Reviewing Policy:**

Affirmed.

**Final Justification:**

I recommend acceptance of this paper. The authors have addressed my concerns during the rebuttal phase and they promise to include additional experiments that I think will strengthen the paper. Overall, I view this paper as providing a useful contribution to the literature.

**Key Questions For Authors:**

In the previous section I have listed a few minor corrections/adjustments that should be made.

In terms of more important questions, here are a few comments:

1. I think that it is absolutely necessary that the authors address any concerns about "correctness" by using dedicated correctness checks for MCMC. For instance, one can use the Geweke test [1] or the "exact invariance test" from [2] (which is a modified version of the former). This can be accompanied with Kolmogorov--Smirnov statistics, etc. I would like to see extensive simulations confirming $\pi$-invariance from this standpoint.

2. The authors need to add more discussions on the limitations of the proposed methodology. It is very exciting to see these results, but there needs to be some grounding so that it doesn't read as being "too good to be true".

[1] Geweke, J. (2004). Getting it right: Joint distribution tests of posterior simulators. Journal of the American Statistical Association, 99(467), 799-804.

[2] Bouchard-Côté, A., Chern, K., Cubranic, D., Hosseini, S., Hume, J., Lepur, M., ... & Sgarbi, G. (2022). Blang: Bayesian declarative modeling of general data structures and inference via algorithms based on distribution continua. Journal of Statistical Software, 103, 1-98.

**Limitations:**

The authors can add more discussion on the limitations of the work. I have commented on this point above.

**Strengths And Weaknesses:**

The submission seems technically sound. I went over a few of the major results and $\pi$-invariance proofs, all of which seem to use standard and well-established procedures. Therefore, I don't question the backbone of the theoretical results. The paper is also reasonably well-structured, although I have left a few comments on areas for improvement at the end of this response.

I would say that the paper is quite significant. Sampling from distributions with difficult (or impossible) to evaluate likelihoods/gradients has been a large burden for many MCMC practitioners. Therefore, any new theory that unifies existing methods and presents a new approach to sampling (e.g., to allow auxiliary variables in both the proposal and acceptance steps) is more than welcome. In terms of originality, the work connects past work and extends it reasonably.

Below are a few more comments that should be addressed by the authors:
- Sometimes you use $\omega$ and sometimes $w$ for the auxiliary variables. Please unify this. (E.g., see the beginning of Section 2.)
- In Proposition 2, "reversible respect to" should be "reversible with respect to"
- Footnote on page 3: "rejectionion" should be "rejection"
- The flow of Section 3.2 is a bit awkward. For instance, the discussion on locally balanced proposals seems to come out of nowhere and much of the content here would really only make sense to readers that are already familiar with the past content.
- In Section 4.3 you refer to Theorem 2 but there is no such theorem. Also, the numbering of lemmas/theorems/propositions is confusing. Are they being manually numbered? Ideally their numbers should be automatically determined by the compiler.
- In Section 5.1 you introduce $\beta$ but define it later in the paragraph.
- What do you mean by "clockwise" MSE in Figure 1?
- Please do a quick pass on the references to ensure proper formatting. For instance, some words are not capitalized: "bayesian" should be "Bayesian", etc. Also, please use an "en" dash (-- in LaTeX) when you have words like Metropolis--Hastings, instead of a hyphen).

---

> ### Author Rebuttal · Authors · 2026-03-30
>
> Thank you very much for your comments and reviews! We apprecaite that you find our paper well-structured. We will fix the typos, formatting and presentation issues you mentioned in our final revision. We address your questions below.
>
> **(1) Correctness Checks**
>
> Thank you for your comment! We agree that dedicated correctness checks are necessary for MCMC methods. In the [anomynous github repo](https://anonymous.4open.science/r/Auxiliary-MCMC-ICML2026-5ECD), we now include a direct correctness study for the truncated Gaussian example, where exact posterior samples can be obtained by running rejection sampling for 1 million steps. We then run each MCMC algorithm under a matched compute budget, discard the first 20% of iterations as burn-in, and compute Kolmogorov--Smirnov statistics against the exact posterior samples. The results are summarized in Table 4. Our method attains KS statistics below 0.05 in all reported dimensions.In contrast, the KS test also reveals the systematic bias of SGLD, with the KS statistic reaching 0.4 in the worst dimension. These results support that our method is sampling from the correct target distribution in this benchmark.
>
> We agree that tests such as the Geweke test and the exact invariance test are also useful. In our tall-data setting, however, applying these tests at the scale of the main experiments would require repeatedly simulating datasets and rerunning posterior sampling many times, which is computationally expensive. Instead, we chose a controlled setting where exact posterior samples are available, since this provides a clean and direct way to assess correctness.
>
> We will add the correctness checks in the revised version. Thanks!
>
> **(2) Limitations:** Good point. Based on our experience, these algorithms seem to have three main limitations. First, they require precomputing several quantities, such as
>  in Poisson-X and
>  in Tuna-X. In practice, these quantities may be unavailable, or only available through loose bounds, which can hurt performance. Second, our algorithm also involves hyperparameter tuning, mainly to choose a good balance between minibatch size and estimator variance. Third, the current design is limited to the independent-data setting. We will also present these limitations more clearly near the end of the paper. Thank you for the helpful suggestions.
>
> **(3) Clock-wise MSE:** We intend to report the MSE of these algorithms as a function of runtime.
>
> **4) Typos, formatting issues, notations:** We will correct them all. Thank you for the careful reading of our paper!

---

> > ### Author Rebuttal · Reviewer_4qKg · 2026-04-02
> >
> > The authors have fully addressed my concerns and I am maintaining my current score.

---

### Decision · Program_Chairs · 2026-04-30

**Decision:**

Accept (spotlight)

**Comment:**

The paper unifies existing MCMC algorithms into common framework and derive new methods using auxiliary variables, showing significant performance gains compared to existing methods. Specifically, the paper provides a principled and exact solution to the long-standing open problem of correcting SGLD's systematic bias using a Metropolis-Hastings step based strictly on minibatches (reviewer XnjG).
All reviewers recommended strong accepts. They found the presentation clear, the result significant (addressing important problems for practitioners), and the framework to be elegant and rigorous.
In responses, the authors provided clearer comparison to baselines, experiments on larger datasets, added correctness checks, and discussed limitations and hyperparameter tuning, addressing reviewer concerns.